# The timescale and direction of influence of a third inferior alternative in human value-learning
Maryam Tohidi-Moghaddam [1,2] ✉ & Konstantinos Tsetsos [1,3] ✉

The way humans and other animals represent the values of alternatives is context-dependent, as it can be distorted by inferior alternatives that are immediately available for choice (immediate context); or that were encountered in previous episodes (temporal context). Yet, the extent to which the immediate and temporal context (co-) shape context-dependent valuation remains unclear. Here, we asked human participants (onsite: $N = 30$, online: $N = 68$) to learn the values associated with three alternatives and explicitly report these values before making binary and ternary choices among the alternatives. We show that context-dependent valuation is evident in the pre-choice value estimates and manifests equally in binary and ternary choices. Accordingly, we conclude that value representations are modulated by the temporal (and not the immediate) context. The direction and across-participants variability of this modulation cannot be captured by extant normalization theories but by a mechanism constructing values through sequential binary comparisons.

Does the way we regard two highly desirable products (3rd generation vs. 2nd generation wireless earbuds) change when the seller introduces a third less desirable product (wired earbuds)? Normative theories posit that the relative preference between two high-value alternatives should be independent of the value and the number of other alternatives that are present in the choice set[1,2]. However, contrary to this premise, empirical findings have shown that the choice between a higher-value (HV) and a lower-value (LV) alternative can be affected by adding a third distractor value (DV) alternative whose value is lower than the lower-value (LV) alternative. This phenomenon is known as the distractor effect[3–6]. The distractor alternative can impact choice quality either in a positive or in a negative manner. In the *positive distractor* effect, the tendency to choose higher-value (HV) over lower-value (LV) alternative increases as the value of the distractor value (DV) increases[3,7,8]. By contrast, in the *negative distractor* effect a higher distractor value (DV) reduces the tendency to choose higher-value (HV) over lower-value (LV) alternative[5,6,9,10].

Distractor effects challenge the independence from irrelevant alternatives principle (IIA)[1,2] and pose constraints on the underlying mechanisms that mediate valuation and choice. Random utility models, that apply noise on veridical (unaffected by the context) representations of choice alternatives, predict either a null distractor effect (i.e., logit models) or a feeble positive distractor effect (i.e., probit models, Fig. 1B, left)[11–13]. However, more sizable distractor effects cannot be captured by random utility models. Instead, stronger distractor effects necessitate mechanisms that distort the value of a certain alternative based on the values of other alternatives encountered in a given context[10,14–18].

In particular, in the range normalization model, each alternative value is divided by the range of all values (i.e., maximum *minus* minimum) in the current context[19–21]. Therefore, when the context features a higher distractor value, it leads to a narrower range or smaller normalization term. This gives rise to a higher distorted value (i.e., subjective value) for both of the two high-value alternatives (Fig. 1A, right), and a greater difference between them (Fig. 1A, right, embedded plot), effectively leading to a positive distractor effect (Fig. 1B, right). By contrast, in the divisive normalization model, each value is normalized by the sum of all values in the context[6]. Thus, a higher distractor value results in a larger normalization term, dwarfing both the values of the two high-value alternatives (Fig. 1A, middle) and their value difference (Fig. 1A, middle, embedded plot). Consequently, this type of distortion gives rise to a negative distractor effect (Fig. 1B, middle).

Despite the recent proliferation of experimental studies and the development of normalization models focusing on context-dependence, definitive conclusions on the underlying mechanisms remain elusive[3,4,6,22–24]. First, there are controversies around the exact direction and robustness of distractor effects. Specifically, both positive[3,8,25] and negative[5,6] distractor effects have been reported using different experimental paradigms ranging from reward-learning in primates to risky and preferential choice paradigms

[1]Department of Neurophysiology and Pathophysiology, University Medical Center Hamburg-Eppendorf, Hamburg, Germany. [2]Department of Psychology and Hamburg Center of Neuroscience, Universität Hamburg, Hamburg, Germany. [3]School of Psychological Science, University of Bristol, Bristol, UK.
✉e-mail: tohidi.mm@gmail.com; k.tsetsos@bristol.ac.uk

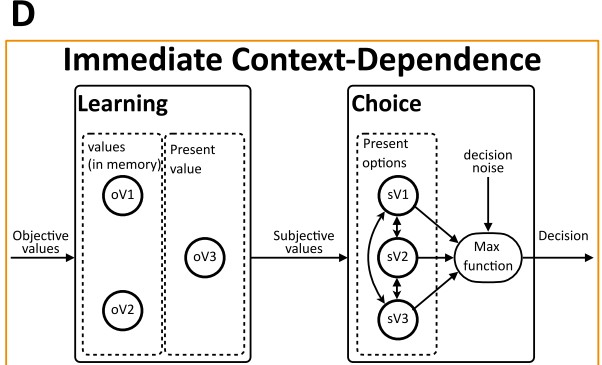
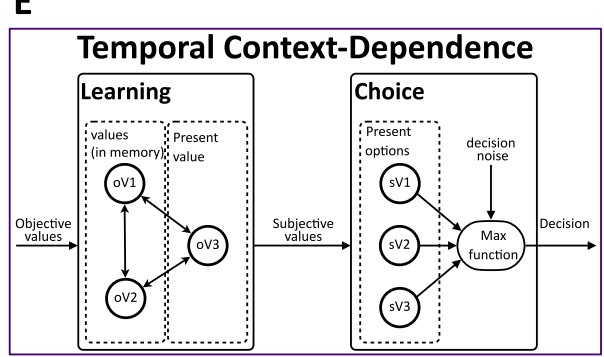

**Fig. 1 | Computational models of distractor effects. A** Value representations for each alternative by probit (left), divisive normalization (middle), and range normalization (right) models. The higher-value (HV) and lower-value (LV) alternatives were kept fixed and the distractor value (DV) changed. As the distractor value increases, the probit model maintains a constant value for two high-value alternatives (HV and LV), a divisive normalization model predicts a decrease of HV and LV values, and a range normalization predicts an increase of HV and LV values. The embedded plots depict the difference between the value representation of HV and

LV as a function of the distractor value. **B** Simulated relative choice of HV over LV in the "immediate-context" and **C** "temporal-context" frameworks. In each subplot, *P(HV over LV)* is calculated separately for ternary (HV, LV, DV) and binary choice trials (HV, LV). **D** Visual depiction of the immediate and **E** temporal-context-dependence frameworks. The two-sided arrows indicate computations leading to context-dependent distortions between alternatives. One-sided arrows indicate the transfer of information to the corresponding node (for further details, see "Model simulations" section).

in humans. Some of these effects failed to replicate[4,25] or changed direction after re-analysis[3,22]. Second, previous studies have not systematically characterized the timescale of context-based distortions. Value representations can be distorted on the basis of the values of alternatives that were previously encountered (temporal context, Fig. 1E)[18,26,27]; and also, on the basis of the values of the alternatives that are immediately available on a given choice trial (immediate context, Fig. 1D)[16,28,29]. It is worth mentioning that some context effects arise in one-shot decisions embedded in non-specific temporal contexts[4,6,8,25]. In these cases, the immediate context account is the only viable one. Similarly, in some learning tasks, the only viable account for context-based distortions is the temporal context[30]. However, in tasks that involve learning and choosing among alternatives within specific temporal contexts (i.e., with each context being characterized by a different distractor value), observed context effects in choice could be attributed to the immediate context, the temporal context, or a combination of both[6,23,30,31]. The key experiments that could measure independently the relative influence of the immediate context and the temporal context have yet to be conducted.

Driven by the above, we here probed distractor effects within a single experimental paradigm that does not tap upon complex cognitive processes evoking alternative interpretations for distractor effects[3,8]. In particular, we used a simple value-learning task previously shown to induce a negative distractor effect in primates[6]. Like in previous studies, to assess the magnitude of the distractor effect, we varied the value of the distractor value (DV) across contexts (experimental mini-blocks). Our design differed from previous studies[5,32] in the following aspects. First, in our task, learning and choice were separated such that the reward values associated with each alternative were independently learned before the choice phase. Second, we interleaved a value estimation task immediately after the learning phase so that we could directly gauge people's subjective values before the onset of the choice phase. That way we could ask whether value distortions already present at the end of the learning epoch, could explain away context effects occurring in the choice phase, consistent with the temporal-context account (Fig. 1C, E). Finally, in the choice phase, we interleaved binary and ternary choice trials. Comparing the relative choice of higher-value (HV) over lower-value (LV) in these two trial types, we could assess the extent to which the immediate availability of the distractor alternative affects choice (Fig. 1B, D, immediate-context account).

To outline our results, in two experiments (onsite and online) we report that value distortions operate on a longer timescale (i.e., the temporal context) while the immediate context had no effect. Specifically, we found that subjective value estimates (elicited in the value estimation phase, interchangeably referred to as estimated values) predicted well the choice rates; and that choice rates were undifferentiated in binary (distractor absent) and ternary (distractor present) trials. Focusing on the pre-choice phase, the higher-value (HV) and lower-value (LV) estimated values were smaller in high-distractor contexts, which is consistent with the divisive normalization model. Although the difference between higher-value (HV) and lower-value (LV) was, on average, marginally smaller in high-distractor contexts (a weak negative distractor effect), which is also consistent with divisive normalization, a large portion of participants showed a positive distractor effect that runs against divisive normalization. Inspired by the decision-by-sampling framework[33,34], we describe a stochastic rank-based mechanism that predicts value reduction in high-distractor contexts while also accounting for the individual variability in the distractor effect (both positive and negative distractor effects across different participants).

## Methods
### Participants
**Experiment 1, onsite.** We recruited 30 participants (22 women, $M = 25.3$ years, $SD = 3.41$) from the internal participant pool in the Institute of Neurophysiology and Pathophysiology of the University Medical Center Hamburg-Eppendorf. This sample size had 56% power to capture Cohen's $d = .4$ corresponding to a small to medium effect size with $\alpha = .05$ using a two-sided $t$-test. This experiment was not preregistered.

Before starting the experiment, all participants gave written informed consent. The study was approved by the local ethics committee of the Hamburg Medical Association. All participants had normal or corrected-to-normal vision, and were right-handed, with the exception of one participant who was left-handed. The experimental session lasted between 3 and 3.5 h. Participants received €30–35 for their participation (hourly net rate = €10). In addition, we included a €10 completion bonus and a maximum of €15 task performance bonus to further incentivize participants to accurately perform the task. The average final task performance bonus was €13.76 (95% CI = [€13.63, €13.90]), which was significantly higher than what would have been expected if participants chose randomly ($t_{(29)} = 41.3$, $p < .001$, $d = 7.54$, 95% CI = [7.17, 7.91]).

**Experiment 2, online.** Sixty-eight (28 self-reported women, 38 self-reported men, and 2 undisclosed, $M = 30.5$ years, $SD = 5.5$) participants were recruited via prolific[35] and completed the experiment in two sessions (each approximately 45 min) within one week. This sample size provided us with the power of 90% to capture the effect size of $d = .4$ with $\alpha = .05$ using a two-sided $t$-test. We did not preregister for this study.

Before starting the experiment, all participants gave written informed consent. The study was approved by the School of Psychological Science Research Ethics Committee at the University of Bristol (approval code: 10270). All participants reported that they have normal or corrected-to-normal vision; and 59 of them were right-handed. Participants received £18 for their participation (hourly net rate = £12). In addition, we included a £2 completion bonus and a maximum of £3 task performance bonus to further incentivize participants to accurately perform the task. The average final task performance bonus was £2.39 (95% CI = [£2.35, £2.43]), which was significantly higher than what would have been expected if participants chose randomly ($t_{(67)} = 20.06$, $p < .001$, $d = 2.43$, 95% CI = [2.19, 2.68]).

### Procedure
**Experiment 1, onsite.** The task was implemented using Psychophysics Toolbox, psychtoolbox-3[36,37], and presented on a 21" monitor (1920 × 1080 px screen resolution and 120-Hz refresh rate). Participants viewed the monitor in a dimly lit room from a 60 cm distance.

**Experiment 2, online.** We used the Gorilla Experiment Builder[38] to create and host our experiment. Participants were recruited through Prolific[35].

In both experiments, participants were instructed to complete 48 mini-blocks of a value-learning task (Fig. 2A). Their task was to collect valuable-colored coins to maximize their overall obtained reward. Each mini-block started with a learning phase where participants learned the association between colored coins and reward values (Fig. 2B). Later, they faced binary and ternary choices in the choice phase, aiming to choose the alternative (colored coins) with the highest reward (Fig. 2D). Following the learning and choice phases we added two estimation phases, in which participants reported their subjective value estimates for each colored coin (Fig. 2C). In each mini-block, the value of all three alternatives (i.e., the context) was kept fixed. We randomly assigned the three alternatives, the higher value (HV), lower value (LV), and distractor value (DV), to three colored circles (red, blue, and yellow); this maps them onto three imaginary coins of different values which are specific to each mini-block. In total, we created four unique contexts by deploying two levels of distractor value, DV = {18, 40}, allowing us to assess the distractor effect, and two magnitude levels for high-value alternatives to insert variability into the task, HV = {55, 50}, LV = HV – 5. The magnitude level (HV equal to 55 or 50) did not yield any significant effects in any of the key analyses. Therefore, for the sake of simplicity, we collapsed the data across the two magnitude levels. Combining these four contexts with a full color-reward permutation, led to 24 unique possible mini-blocks (six color-reward permutations times four unique value contexts). These mini-blocks were repeated twice, in a total of 48 mini-blocks. In half of the mini-blocks, we did not provide any reward feedback in the choice phase ("No-Feedback"

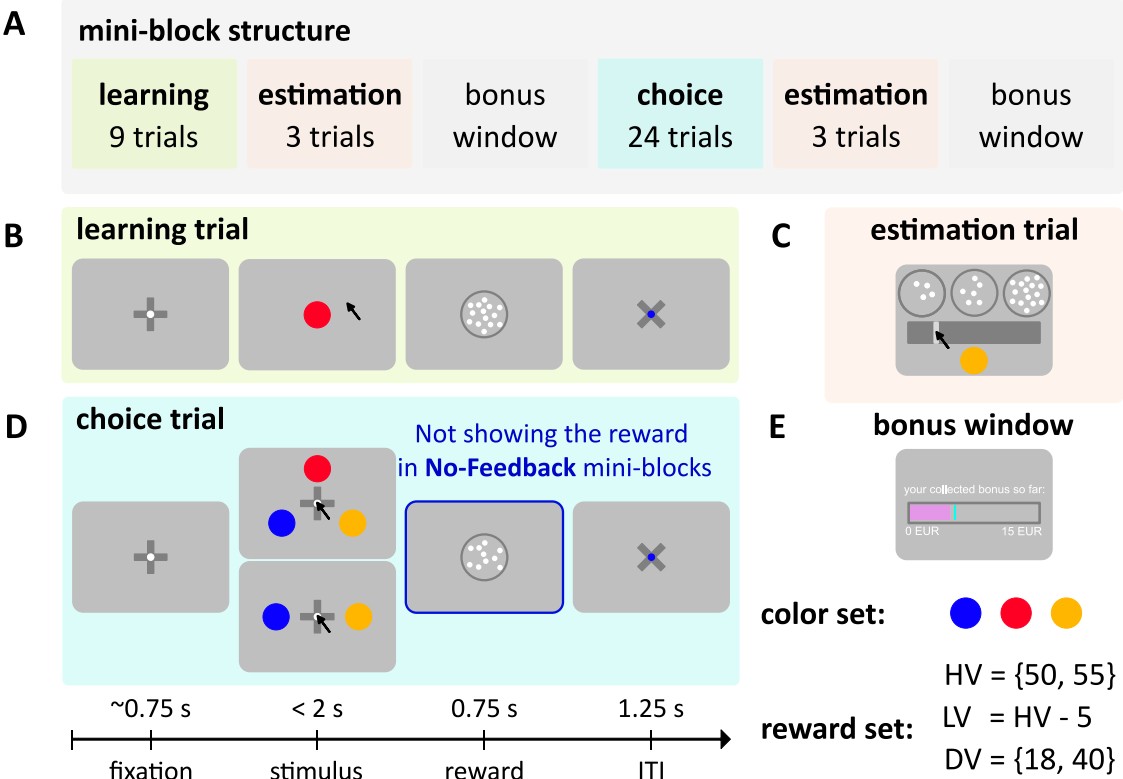

**Fig. 2 | Human value-learning task. A** Mini-block timeline (from left to right), each participant completed 48 mini-blocks of the color-reward association task. They were instructed to maximize their total obtained reward by first learning the color-reward association during the learning phase of each mini-block and then choosing the alternative with the highest reward in the subsequent choice phase. Each mini-block started with the learning phase where participants sequentially observed the values associated with three alternatives (imaginary colored coins, look at the color set) in a given context. Each context included a unique HV, LV, and DV alternative (c.f. reward set). Following the learning phase, each mini-block continued with the first estimation phase where participants reported their subjective value estimate of each mini-block. After the estimation phase, the bonus window shortly appeared before the choice phase began. In the choice phase, participants had to choose the alternative that yielded the largest reward. The choice phase was followed by the second estimation phase and each mini-block ended with the presentation of the final bonus window. **B** The learning phase consisted of 9 learning trials (3 per each alternative). Each trial began with the fixation cue for a random duration, and after that one of the alternatives appeared on the screen. Participants were asked to click on the alternative within 2 s to get the associated reward (i.e., represented by the number of a set of moving dots). **C** The Estimation phase involved 3 trials (1 per each alternative). On each trial, an alternative was presented below a slider bar and participants were asked to move the slider to indicate the number of dots (i.e., reward value) associated with that alternative. **D** The choice phase included 24 trials (12 ternary and 12 binary). Like learning trials, each choice trial initiated with the fixation cue for a random delay, and after that, 2 or 3 alternatives presented on the screen. Participants were asked to click on their preferred alternative within 2 s to see the collected reward. In "No-Feedback" mini-blocks, the choice was immediately followed by the inter-trial-interval (ITI) window, and the collected reward was not presented. **E** The bonus window, presented after each estimation phase, revealed the participant's collected monetary bonus in a pink bar and the possible maximum collected bonus with a cyan line so far. All panels are based on Experiment 1, the design of both experiments was identical, the only minor change in the online experiment was the use of keyboard instead of mouse clicks to record participants' responses at each phase. Note that the panels are a schematic (cartoon) of the trials, for exact details of the stimuli and reward, see "Methods" section.

mini-blocks) while in the other half, we showed the reward associated with each choice ("Feedback" mini-blocks). We instructed participants that "Feedback" and "No-Feedback" mini-blocks were randomly interleaved and that there was no way to know which mini-block they were performing prior to the choice phase. This ensured that the participants paid attention to the learning phase equally in "Feedback" and "No-Feedback" mini-blocks, enabling us to attribute any behavioral disparities across the two conditions to the choice phase. In Experiment 1, all participants received verbal instructions about the task structure and commenced the main experiment after completing two practice mini-blocks. In Experiment 2, however, we were unable to provide verbal instructions; instead, we provided participants with detailed written instructions before starting the experiment followed by six comprehension questions. We only allowed the participants who correctly answered all six comprehension questions about the instructions to participate in the experiment.

**Stimuli**

**Alternatives.** The stimuli were three colored filled circles (diameter size = 1.5 visual degrees according to Experiment 1). The colors were yellow = [255 250 100], red = [255 90 90], and blue = [100 153 255]. Stimuli were presented on a gray background, [127.5 127.5 127.5].

**Reward.** The presented reward was a set of moving dots (diameter dot = 0.15 visual degrees according to Experiment 1) framed in a circle (frame thickness = 0.075 visual degrees according to Experiment 1, frame color = [51 51 51]). The number of dots was equal to the associated value of the chosen alternative (see the value sets mentioned in the main task section). The color of each dot was grayscale ([127.5 127.5 127.5]) and its contrast was drawn from a uniform distribution: [0.6 1]. The reward presentation duration in both learning and "Feedback" choice trials was 0.75 s. We presented the dots in 10 consecutive frames (frame duration = 75 ms) and induced motion by randomly assigning each dot to a new location in each frame. To make the human reward feedback as similar as possible to the monkeys' juice delivery in the original study[6], we needed to introduce some degree of "internal" uncertainty into the reward perception. Thus, instead of showing the dots statically on the screen, we presented them in a dynamically moving format, to induce uncertainty at the perceptual level.

## Mini-block structure

**Learning phase.** Participants completed 9 randomly presented learning trials (three per each alternative). Each trial started with the presentation of a fixation cue for a random short time (randomly drawn from a truncated exponential distribution: [0.5 1] and mean = 0.75 s). Then the stimulus, one colored coin, was presented at the center of the screen for a maximum of 2 s. Participants were asked to learn the value of the stimulus within 2 s and collect (by clicking on that in Experiment 1 and pressing the "SPACE" bar in Experiment 2) its associated reward. If they failed to do so, they missed the reward and saw a "Missed response" message on the screen; in Experiment 1, once the stimulus appeared on the screen, the mouse pointer appeared on a random position of an imaginary circle whose center aligned with the stimulus (diameter = 2 visual degrees). Immediately after their response, participants could see a centrally presented cloud of moving dots representing the reward associated with the stimulus. The reward was presented for 0.75 s and then followed by a gray screen with the fixation point on the center as inter-trial-interval (ITI) (Experiment 1: ITI = 1.25 s, and Experiment 2: ITI = 0.25 s).

**Estimation phase.** We had 3 estimation trials, one for each alternative, presented in random order. In each estimation trial, we presented a target alternative (shown at the bottom-center of the bar) and asked participants to report the estimated value of the alternative corresponding to. They did so by first moving a slider on a bar within 10 s to find the best estimation and then pressing the "SPACE" bar to confirm their estimation. After 10 s, the current trial was automatically terminated and the next estimation trial was initiated. At the top of the two ends of the bar, we statically displayed the minimum (15 dots) and maximum (85 dots) possible number of dots, which we encouraged participants to use as reference. As participants adjusted the slider, they could instantly observe the number of moving dots within a set displayed at the top center in Experiment 1 and as a numerical value in Experiment 2, both changing in real-time.

**Choice phase.** The choice phase included 12 ternary and 12 binary choice trials which were presented in random order. We had 3 combinations of binary trials {(HV, LV), (HV, DV), (LV, DV)} and presented each 4 times. The mapping between the location of the alternatives, (left, top, right) in ternary trials and (left, right) in binary trials, and the alternatives was counterbalanced. As in the learning trials, each trial began with the presentation of a fixation cue for a short delay (randomly drawn from a truncated exponential distribution: [0.5 1] and mean = 0.75 s). Then the stimuli appeared on the screen for a maximum of 2 s. The stimuli were equidistant from the center (eccentricity = 2 visual degrees). Participants were asked to report (by clicking on that in Experiment 1 and pressing the associated arrow key in Experiment 2) their preferred stimulus within 2 s to collect the associated reward for their choice. Failure to respond within the deadline resulted in a "Missed response" message appearing on the screen; in Experiment 1, on each choice trial the mouse pointer was initiated at the center of the screen. In "Feedback" mini-blocks, the reward was presented at the center of the screen immediately after a choice was made. The reward stayed on the screen for 0.75 s, followed by ITI (Experiment 1: ITI = 1.25 s, and Experiment 2: ITI = 0.25 s). In "No-Feedback" mini-blocks, a choice was directly followed by the ITI window.

**Bonus window.** We presented a bonus window twice, at the end of each estimation phase. In Experiment 1, this window illustrated the cumulative bonus earned from the start of the task until that moment, represented by a pink bar whose length visually reflected the accumulated bonus. A cyan line indicated the maximum possible bonus attainable. Consequently, the space between the end of the pink bar and the cyan represented the bonus participants had missed. In Experiment 2, we used two different bars, one showing the collected reward so far in green and the other one presenting the missed reward in red. The length of each bar

indicated the maximum bonus they could either achieve or miss in the experiment.

## Analyses

Across different analyses, we used one-sample $t$-tests, ANOVA, and Pearson correlation. In the correlation analyses, we first obtained a correlation coefficient for each participant and then compared the correlation coefficients of the group against zero. All analyses were done using Matlab[39]. For the interpretation of null effects, we used JASP[40] and its default setting to calculate the corresponding Bayes factor ($BF_{01}$) in favor of the null hypothesis.

To quantify the distractor effect in the choice phase, we compared the relative choice (HV over LV) in the two high-distractor and low-distractor contexts. The relative choice is quantified as:

$$p(HV \text{ over } LV) = \frac{p(HV)}{p(HV) + p(LV)} \tag{1}$$

where $p(HV)$ and $p(LV)$ indicate the choice rate of the higher-value and lower-value alternatives. The distractor effect (DE) is quantified as:

$$DE = p(HV \text{ over } LV | high_D) - p(HV \text{ over } LV | low_D) \tag{2}$$

Thus, $DE > 0$, $DE < 0$, and $DE = 0$, indicate a positive, a negative, and a null distractor effect respectively. We only analyzed trials in which participants responded within the 2-s deadline in the learning and choice phases and within the 10-s deadline in the estimation phases.

**Choice prediction using subjective values.** To convert the estimated values into pseudo-choices, we applied an *argmax* selection rule (like the one used here[30]) on each ternary trial as:

$$p(a) = \begin{cases} 1 \text{ if } SV(a) > SV(b) \text{ and } SV(a) > SV(c) \\ .5 \text{ if } SV(a) > SV(b) \text{ and } SV(a) = SV(c) \\ .3 \text{ if } SV(a) = SV(b) = SV(c) \\ 0 \text{ if } SV(a) < SV(b) \text{ or } SV(a) < SV(c) \end{cases} \tag{3}$$

and on each binary trial as:

$$p(a) = \begin{cases} 1 \text{ if } SV(a) > SV(b) \\ .5 \text{ if } SV(a) = SV(b) \\ 0 \text{ if } SV(a) < SV(b) \end{cases} \tag{4}$$

where $SV(x)$ is the value estimate for alternative $x$ and $p(x)$ is the predicted choice probability of that alternative in each trial.

## Computational models

To model the choice behavior, objective values ($OV_i$) were transformed into "subjective" values ($SV_i$), with the transformation type differing across models. Transformed values ($SV_i$) were converted to choice probabilities by applying a probit function (using a numerical approach, Eq. 5). For each alternative $i$ the choice probability $p(i)$ was calculated as:

$$p(i) = \int_{-\infty}^{\infty} \left( f\left(x; SV_i, \sigma_f\right) \cdot \prod_{j \neq i}^{n-1} F\left(x; SV_j, \sigma_f\right) \right) dx \tag{5}$$

where $f$ indicates the probability density function and $F$ is the cumulative distribution function. Here we assumed that all represented subjective values are normally distributed ($N(SV, \sigma_f^2)$) and $n$ is the number of alternatives. We used this range: [eps 100] for $\sigma_f$ to fit the model.

We used four computational models with each describing an alternative mechanism mapping objective values into their subjective (transformed) counterparts.

**Probit model (Pr).** This model assumes that the subjective values should be unaffected by the choice-set context. Thus, there is no transformation in this model and the subjective values are encoded as equal to the objective values.

$$SV_i = OV_i \qquad (6)$$

**Divisive normalization model (DN).** The subjective value of each alternative is normalized by the sum of all alternatives.

$$SV_i = k * \frac{OV_i}{\sigma + w * \sum_{n=1}^{3} OV_n} \qquad (7)$$

where $k$, $\sigma$, and $w$ are gain, semi-saturation, and weight terms. To fit the model we constrained parameters in these ranges, respectively: [1 200], [1 200], and [0 1].

**Range normalization model (RN).** The subjective value of each alternative is normalized by the range of values in a given context.

$$SV_i = \frac{OV_i}{1 + w * (\max(OV's) - \min(OV's))} \qquad (8)$$

where $OV's$ refers to a vector consisting of all alternative values in the context and $w$ is a weight term. To fit the model, we constrained $w$ between [0 1].

**Rank-based model (RB).** This model implements relative value coding using a series of memory-based binary comparisons across all possible pairs of alternatives in the context. The subjective value of each alternative is the total number of times that this alternative was the winner across all memory-based binary comparisons.

$$SV_i = k_{ij} * \sum_{j \neq i}^{n-1} p_{(i>j)} \qquad (9)$$

where $k_{ij}$ represents the total number of binary comparisons, constrained in the [1 200] range. $p_{(i>j)}$ is the probability that alternative $i$ is deemed as better than alternative $j$ in a given binary comparison. We derived this probability using a softmax function:

$$p_{(i>j)} = \frac{\exp\left(\frac{OV_i}{\tau}\right)}{\exp\left(\frac{OV_i}{\tau}\right) + \exp\left(\frac{OV_j}{\tau}\right)} \qquad (10)$$

where $\tau$ is the temperature parameter, which was constrained in the [1 100] range when fitting the model.

**Model simulations.** The model predictions shown in Fig. 1 were generated using numerical approximations (see Eq. S3). In all models, we used the same value sets: HV = 50, LV = 45, and DV = {0, 7, 13, 20, 27, 33, 40}. The parameters for each model were fixed as follows: Pr: $\sigma_f^2 = 30.25$, DN: $\sigma_f^2 = 9$, $k = 100$, $\sigma = 50$ and $w = 1$, and RN: $\sigma_f^2 = 0.25$, and $w = 1$. In Fig. 5D, E, the parameters to simulate the RB model in equal binary comparison were: $\sigma_f^2 = 36$, $\tau = 20$, $k_{HVvsLV} = k_{HVvsDV} = k_{LVvsDV} = 40$, and in unequal binary comparison were: $\sigma_f^2 = 400$, $\tau = 20$, $k_{HVvsLV} = 60$, $k_{HVvsDV} = 40$, $k_{LVvsDV} = 20$. In Fig. 5F, we used the same value sets as Fig. 5D, E: HV = 50, LV = 45, but only two levels of distractors DV = {18, 40} to be able to calculate the distractor effect (Eq. 2) and the parameters were: $\sigma_f^2 = 100$, $\tau = 20$, $k_{HVvsLV} = 20–60$, $k_{HVvsDV} = 20–60$, $k_{LVvsDV} = 20–60$.

**Model fitting procedure.** We fitted the models by minimizing the negative log-likelihood (NLL) summed over all choice trials (including all ternary and binary trials)[41]. For each participant, we fitted each model once on the entire set of choice trials, including both "Feedback" and

"No-Feedback" mini-blocks (~1152 choice trials). We optimized the parameters using Matlab's *fmincon.m* function (MaxFunEvals = 5000; MaxIter = 5000; TolFun = 1e-20; TolX = 1e-20). To avoid local optima, we refitted each model for each participant 10 times using a grid of randomly generated starting values for the free parameters.

**Model selection.** We used a fixed-effects and a random-effects model comparison. We used the Bayesian information criterion (BIC) to compare different models. The BIC is quantified as below:

$$BIC = k * \ln(n) - 2 * \ln(L) \qquad (11)$$

where $L$ is the likelihood value obtained from the model fit to all choice data, $k$ is the number of free parameters in each model, and $n$ is the number of trials we used to fit the model[42]. We computed the BIC score for each participant and each model. Then for each participant the model with the lowest BIC was marked as the best-individualized model. Finally, we reported the $\Delta BIC$ of each model relative to the best-individualized model (Fig. 6B).

We also fitted the models using a 6-fold cross-validation procedure. To do so, for each participant, we split the trials into 6 parts (i.e., folds) and then fitted each model to a "training" set (comprising five random folds). We used the best-fitted parameters of the "training" set to calculate the LL summed across trials in the left-out "test" fold. We repeated this process over test folds (6 times) and the final cross-validated LL was computed as the mean LL across cross-validated folds. We then used this mean LL to calculate each model's posterior frequency and protected exceedance probability (i.e., the probability corrected for the chance level that a model is more likely than any others) using the variational Bayesian analysis (VBA) toolbox[43,44].

**Reporting summary**

Further information on research design is available in the Nature Portfolio Reporting Summary linked to this article.

## Results
### Basic performance

During the learning phase (Fig. 2B), participants were sequentially exposed to the value associated with each alternative. The value was shown by a cloud of moving dots, with the number of dots representing the reward magnitude (i.e., number of points) associated with the coin (see "Methods" section). Participants completed 99.91% (95% CI = [99.84%, 99.97%]) of all learning trials with an average reaction time of 623.66 milliseconds (ms) (95% CI = [593.56 ms, 653.76 ms]) in Experiment 1, and 99.3% (95% CI = [98.94%, 99.66%]) of all learning trials with an average reaction time of 411.92 ms (95% CI = [386.66 ms, 437.18 ms]) in Experiment 2, indicating that they properly engaged with the learning phase. We found no statistically significant evidence for differences between the reaction times across learning trials that involved the higher-value (HV), lower-value (LV), and distractor value (DV) alternatives across different contexts in either experiment (2-way ANOVA on $log$(RT) in Experiment 1: alternatives: $F_{(2, 174)} = 0.004$, $p = 0.99$, partial $\eta^2 < 0.01$, 95% CI = [0.00, 0.02], contexts: $F_{(1, 174)} = 0.15$, $p = 0.69$, partial $\eta^2 < 0.01$, 95% CI = [0.00, 0.03], interaction: $F_{(2, 174)} = 0.02$, $p = 0.98$, partial $\eta^2 = 0.00$, 95% CI = [0.00, 0.02]; and in Experiment 2: alternatives: $F_{(2, 402)} = 0.03$, $p = 0.97$, partial $\eta^2 = 0.00$, 95% CI = [0.00, 0.01], contexts: $F_{(1, 402)} = 0.06$, $p = 0.81$, partial $\eta^2 < 0.01$, 95% CI = [0.00, 0.01], interaction: $F_{(2, 402)} = 0.04$, $p = 0.97$, partial $\eta^2 < 0.01$, 95% CI = [0.00, 0.01]).

In the choice phase (Fig. 2D), participants made 12 ternary and 12 binary choices (four trials per each possible binary pair) by reporting their preferred alternative within a maximum of two seconds. We included all three possible binary pairs (HV vs. LV, HV vs. DV, LV vs. DV) to keep the presentation rate of all alternatives equal and to avoid introducing any implicit bias toward the two high-value alternatives. Participants completed 99.73% (95% CI = [99.62%, 99.84%]) of all choice trials with an average

reaction time of 729.89 ms (95% CI = [695.14 ms, 764.65 ms]) in Experiment 1, and 99.34% (95% CI = [99.08%, 99.60%]) of choice trials with an average reaction time of 614.50 ms (95% CI = [581.55 ms, 647.45 ms]) in Experiment 2. Participants chose the best alternative significantly higher than the chance level in both ternary and binary trials (Experiment 1: accuracy in ternary = 81.63% (95% CI = [78.49%, 84.77%]), $t(29) = 30.17$, $p < 0.001$, $d = 5.51$, 95% CI = [5.13, 5.88]), and in binary = 90.27% (95% CI = [88.29%, 92.25%]), $t(29) = 39.86$, $p < 0.001$, $d = 7.28$, 95% CI = [6.90, 7.65]; Experiment 2: accuracy in ternary = 70.08% (95% CI = [66.81%, 73.35%]), $t(67) = 22.03$, $p < 0.001$, $d = 2.67$, 95% CI = [2.43, 2.91], and in binary = 81.23% (95% CI = [78.66%, 83.81%], $t(67) = 23.80$, $p < 0.001$, $d = 2.89$, 95% CI = [2.64, 3.13]; Fig. S1 in Supplemental Information). The average reaction time in binary trials was significantly higher than in ternary trials only in Experiment 1, with no statistically significant evidence for change across different contexts in both experiments (2-way ANOVA on $log$(RT) in Experiment 1: choice type: $F(1, 116) = 11.40$, $p < 0.001$, partial $\eta^2 = 0.09$, 95% CI = [0.02, 0.21], contexts: $F(1, 116) = 0.20$, $p = 0.65$, partial $\eta^2 < 0.01$, 95% CI = [0.00, 0.05], interaction: $F(1, 116) = 0.03$, $p = 0.87$, partial $\eta^2 < 0.01$, 95% CI = [0.00, 0.04]; and in Experiment 2: choice type: $F(1, 268) = 2.97$, $p = 0.09$, partial $\eta^2 = 0.01$, 95% CI = [0.00, 0.05], contexts: $F(1, 268) = 0.93$, $p = 0.34$, partial $\eta^2 < 0.01$, 95% CI = [0.00, 0.03], interaction: $F(1, 268) = 0.01$, $p = 0.92$, partial $\eta^2 < 0.01$, 95% CI = [0.00, 0.02]; Fig. S1 in Supplemental Information). This seemingly counterintuitive finding might be attributed to the greater uncertainty linked with the identities of the alternatives in binary trials.

Moreover, in half of the mini-blocks, we withheld reward feedback in the choice phase to intercept further learning from feedback after the value estimation phase. Withholding feedback could help us better quantify immediate-context effects. We found no statistically significant evidence that receiving feedback improved choice accuracy in Experiment 1 (choice accuracy pooled across both ternary and binary: in "Feedback", 89.00% (95% CI = [86.25%, 91.76%]) and in "No-Feedback", 87.22% (95% CI = [85.02%, 86.42%]) mini-blocks; two-sided $t$-test: $t(29) = 1.54$, $p = 0.13$, $d = 0.28$, 95% CI = [−0.09, 0.65]). However, in Experiment 2, accuracy was significantly higher in the Feedback condition (choice accuracy pooled across both ternary and binary: in "Feedback", 79.69% (95% CI = [77.14%, 82.25%]) and in "No-Feedback", 77.19% (95% CI = [74.11%, 80.27%]) mini-blocks; two-sided $t$-test: $t(67) = 3.20$, $p < 0.01$, $d = 0.39$, 95% CI = [0.15, 0.63]). There was no statistically significant difference in the average reaction time between receiving and not receiving feedback in both experiments (reaction time pooled across both ternary and binary in Experiment 1: "Feedback", 730.46 ms (95% CI = [696.85 ms, 764.07 ms]) and "No-Feedback", 729.36 ms (95% CI = [691.98 ms, 766.75 ms], two-sided $t$-test on $log$(RT): $t(29) = 0.39$, $p = 0.70$, $d = 0.07$, 95% CI = [−0.30, 0.44]); in Experiment 2: "Feedback", 609.23 ms (95% CI = [576.25 ms, 642.29 ms]), and "No-Feedback", 619.84 ms (95% CI = [585.92 ms, 653.76 ms]) mini-blocks; two-sided $t$-test on $log$(RT): $t(67) = −1.40$, $p = 0.17$, $d = −0.17$, 95% CI = [−0.41, 0.07]). Overall, the choice rate (i.e., the number of times one alternative is chosen *divided by* the number of times that alternative is presented) of each value was consistent with value-guided decision-making (Experiment 1: choice rate of (HV) = 84.83%, (95% CI = [82.17%, 87.49%]), (LV) = 31.80%, (95% CI = [29.94%, 33.65%]), and (DV) = 3.32%, (95% CI = [2.08%, 4.56%]); Experiment 2: choice rate of (HV) = 74.31%, (95% CI = [71.33%, 77.28%]), (LV) = 35.85%, (95% CI = [34.59%, 37.10%]), and (DV) = 9.80%, (95% CI = [7.63%, 11.98%]), pooled across both "Feedback" and "No-Feedback" mini-blocks).

In the estimation phase (Fig. 2C), participants were asked to report their subjectively learned value of each alternative by moving a slider on a bar within 10 s. Participants completed 99.70% (95% CI = [99.51%, 99.89%]) of all estimation trials across both phases with an average reaction time of 3.24 s (95% CI = [2.95 s, 3.52 s]) in Experiment 1, and 98.29% (95% CI = [97.32%, 99.27%]) of them with an average reaction time of 3.31 s (95% CI = [2.93 s, 3.69 s]) in Experiment 2. In both experiments, the average estimated values (i.e., subjective values) correlated strongly with the actual values in estimation phases, (Pearson correlation in Experiment 1, First estimation: $r = 0.98$ (95% CI = [0.98, 0.99]), two-sided $t$-test on correlation coefficients: $t(29) = 334.17$, $p < 0.001$, $d = 61.01$, 95% CI = [60.64, 61.39], Second estimation: $r = 0.98$ (95% CI = [0.97, 0.99]), two-sided $t$-test on correlation coefficients: $t(29) = 251.69$, $p < 0.001$, $d = 45.95$, 95% CI = [45.58, 46.32]; and in Experiment 2, First estimation: $r = 0.97$ (95% CI = [0.96, 0.98]), two-sided $t$-test on correlation coefficients: $t(67) = 230.94$, $p < 0.001$, $d = 28.01$, 95% CI = [27.76, 28.25], Second estimation: $r = 0.96$ (95% CI = [0.95, 0.98]), two-sided $t$-test on correlation coefficients: $t(67) = 139.36$, $p < 0.001$, $d = 16.90$, 95% CI = [16.66, 17.14]). All analyses above indicate that participants engaged well with the estimation and choice tasks.

### Context-dependent distortion following learning

Having ensured that participants complied with the task instructions, we next asked how value estimates (i.e., subjective values) varied as a function of the distractor value. Here, we analyzed the estimation data pooled across both "Feedback" and "No-Feedback" mini-blocks, as this manipulation was announced to the participants at the end of the estimation phase (see "Methods" section). We only focused on the results from the first estimation because these were not influenced by any behavior or feedback received during the choice phase. Figure 3A, D shows the average value estimates for each alternative across contexts. In both experiments, in the first estimation, there was a statistically significant decrease of the estimated values of the two high-value alternatives in in the high-distractor context relative to the low-distractor context (two-sided $t$-test in Experiment 1: HV: $t(29) = 5.33$, $p < 0.001$, $d = 0.97$, 95% CI = [0.60, 1.35], LV: $t(29) = 5.38$, $p < 0.001$, $d = 0.98$, 95% CI = [0.61, 1.36]; in Experiment 2: HV: $t(67) = 4.64$, $p < 0.001$, $d = 0.56$, 95% CI = [0.32, 0.80], LV: $t(67) = 2.14$, $p = 0.036$, $d = 0.26$, 95% CI = [0.02, 0.50]; and pooled across both experiments; HV: $t(97) = 6.55$, $p < 0.001$, $d = 0.66$, 95% CI = [0.46, 0.86], LV: $t(97) = 4.13$, $p < 0.001$, $d = 0.42$, 95% CI = [0.22, 0.62]). This effect indicates that distractor-induced distortions in subjective value already occurred during the learning epoch.

At first glance, this pattern appears consistent with the divisive normalization prediction and runs against the range normalization prediction (Fig. 1A). Divisive normalization also predicts that the value difference between the higher-value (HV) and lower-value (LV) alternatives decreases as the distractor value (DV) increases (embedded plot in Fig. 1A, middle). Figure 3E shows that the value difference between the higher-value (HV) and lower-value (LV) alternatives is in line with the prediction of divisive normalization in Experiment 2 (two-sided $t$-test on "$HV – LV$" between the contexts with low and high distractors, First estimation: $t(67) = 2.27$, $p = 0.026$, $d = 0.28$, 95% CI = [0.03, 0.52]), but Fig. 3B shows that this is not the case in Experiment 1 (two-sided $t$-test on "$HV – LV$" between the contexts with low and high distractors, First estimation: $t(29) = −0.006$, $p = 0.99$, $d = −0.001$, 95% CI = [−0.37, 0.37]). The latter null finding (Bayesian $t$-test on "$HV – LV$" between the contexts with low and high distractors, $BF_{01} = 5.14$) in the value difference is more consistent with a probit model, which however does not predict a global reduction in value estimates (embedded plot in Fig. 1A, left). The difference in the "estimation" distractor effect between the two experiments could presumably be attributed to variations in the demographics of the online and onsite cohorts; or to the overall lower engagement and attention in online studies due to the different environment between lab-based and online studies (evidenced by small but consistent accuracy differences: two-sided $t$-test on overall choice accuracy of best alternative between the two experiments: $t(97) = 4.33$, $p < 0.001$, $d = 0.95$, 95% CI [0.52, 1.39]).

Thus, in two experiments, increasing the distractor value consistently reduced the estimated values of the two high-valued alternatives, while its effect on their difference was marginally negative (two-sided $t$-test on "$HV – LV$" between the contexts with low and high distractors, aggregated across the two experiments; First estimation: $t(97) = 2.04$, $p = 0.044$, $d = 0.21$, 95% CI = [0.01, 0.41]). We return to the interpretation of these results in a later section.

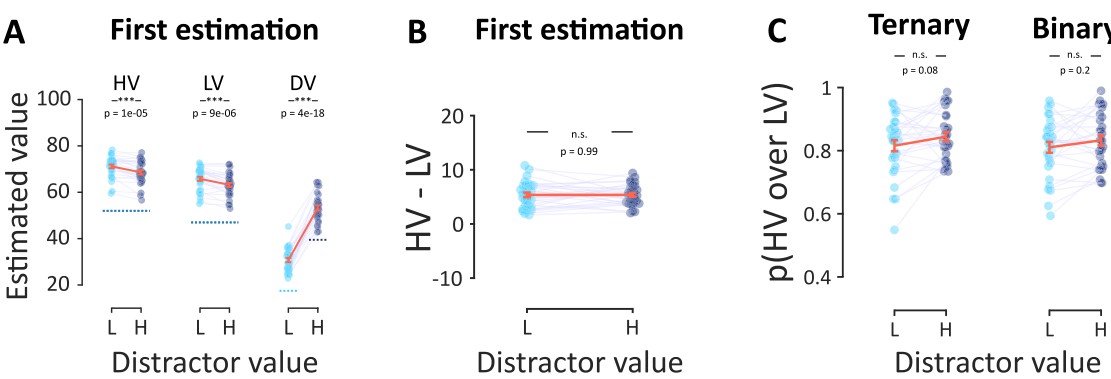

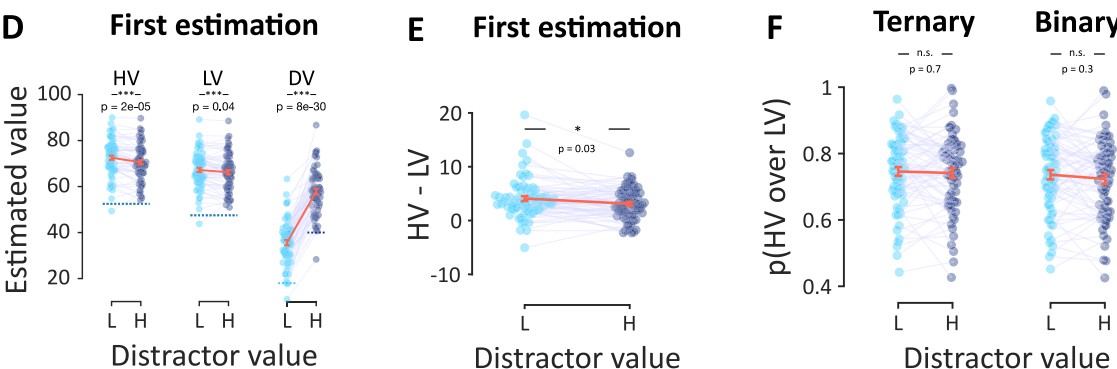

**Fig. 3 | Context dependence in estimated value and choice. A**, **D** Mean estimated value of each alternative as a function of the distractor value. Dashed lines indicate the actual values. **B**, **E** The difference in estimated values between the two high-value alternatives, shown in the two distractor contexts. **C**, **F** Relative choice of HV over LV in the choice phase as a function of the distractor value in ternary trials and binary (HV, LV) trials. In all panels, data were pooled across "Feedback" and "No-Feedback" mini-blocks, the orange error bars indicate standard errors of the mean across participants, and the colored dots indicate individual participants.

## No distractor effect in the choice phase

We next sought to assess whether the choice data revealed a distractor effect. The presence of both binary and ternary trials in the choice phase allowed us to probe the origin of a potential distractor effect by focusing on the relative choice share between the two high-value alternatives (here denoted as $p(HV$ $over LV)$, Eq. 1) and on how this quantity varies across contexts (low- vs. high-distractor) and trial-type (binary vs. ternary). Here, the data is pooled across both "Feedback" and "No-Feedback" mini-blocks, given that there was no statistically significant evidence for an interaction in relative choice share between these two mini-block types (2-way ANOVA on $p(HV$ $over$ $LV)$, in Experiment 1: contexts: $F(1, 116) = 1.88$, $p = 0.17$, partial $\eta^2 = 0.02$, 95% CI = [0.00, 0.09], block type: $F(1, 116) = 4.42$, $p = 0.04$, partial $\eta^2 = 0.04$, 95% CI = [0.00, 0.13], interaction: $F(1, 116) = 0.09$, $p = 0.76$, partial $\eta^2 < 0.01$, 95% CI = [0.00, 0.05]; and in Experiment 2: contexts: $F(1, 268) = 0.16$, $p = 0.69$, partial $\eta^2 < 0.01$, 95% CI = [0.00, 0.02], block type: $F(1, 268) = 1.33$, $p = 0.25$, partial $\eta^2 < 0.01$, 95% CI = [0.00, 0.04], interaction: $F(1, 268) = 3.11$, $p = 0.08$, partial $\eta^2 = 0.01$, 95% CI = [0.00, 0.05]).

Specifically, a main effect of "context", indicating that the relative choice share changes under high-distractor values, would reveal a non-specific distractor effect. Depending on the influence of "trial-type" this effect could be attributed to a purely temporal-context account (no "context" x "trial-type" interaction), a pure immediate-context account (significant interaction, with the distractor effect being present in ternary trials only), or a combination of the two accounts (with the distractor effect in binary trials capturing the temporal influence and the distractor effect in ternary trials the immediate-context influence).

Analysis of the relative choice share across context and trial-type did not provide statistically significant evidence for a "context" main effect, which is necessary to establish a distractor effect (Fig. 3C, F, 2-way ANOVA on $p(HV$ $over LV)$, in Experiment 1: context: $F(1, 116) = 2.5$, $p = 0.12$, partial $\eta^2 = 0.02$, 95% CI = [0.00, 0.10], trial-type: $F(1, 116) = 0.27$, $p = 0.61$, partial $\eta^2 < 0.01$, 95% CI = [0.00, 0.05], interaction: $F(1, 116) = 0.04$, $p = 0.83$, partial $\eta^2 < 0.01$, 95% CI = [0.00, 0.04]; and in Experiment 2: context: $F(1, 268) = 0.41$, $p = 0.52$, partial $\eta^2 < 0.01$, 95% CI = [0.00, 0.03], trial-type: $F(1, 268) = 0.96$, $p = 0.33$, partial $\eta^2 < 0.01$, 95% CI = [0.00, 0.03], interaction: $F(1, 268) = 0.08$, $p = 0.77$, partial $\eta^2 < 0.01$, 95% CI = [0.00, 0.02]).

Further, the non-significant "trial-type" effect (Bayesian ANOVA with $P(M) = 0.2$ for all models on $p(HV$ $over LV)$, in Experiment 1: null model: $BF_{01} = 1.00$, context model: $BF_{01} = 1.65$, trial-type model: $BF_{01} = 4.56$, a full model with interaction: $BF_{01} = 28.69$; and in Experiment 2: null model: $BF_{01} = 1.00$, context model: $BF_{01} = 6.19$, trial-type model: $BF_{01} = 4.74$, full model: $BF_{01} = 152.38$) suggests that the relative choice share is indistinguishable between binary and ternary trials. This means that there was no basic immediate-context value modulation, where the relative choice share would change solely due to the addition of a third alternative in the choice set, regardless of its value. The binary and ternary distractor effects (Eq. 2) were strongly correlated across participants (Pearson correlation, in Experiment 1: $r_{(30)} = 0.90$, $p < 0.001$, 95% CI = [0.80, 0.95]; and in Experiment 2: $r_{(68)} = 0.92$, $p < 0.001$, 95% CI = [0.87, 0.95], Fig. S1 in Supplemental Information) further corroborating the lack of immediate-context value modulation.

## Experiment 1: onsite N = 30

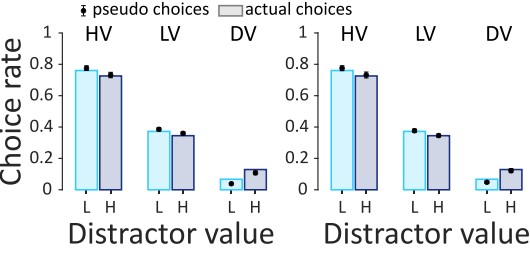

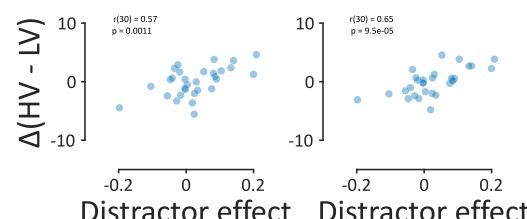

## Experiment 2: online N = 68

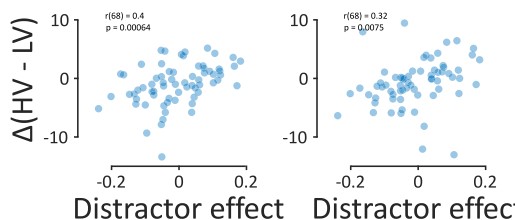

**Fig. 4 | Value representation comparison between the estimation and choice phases. A, C** Choice rates (bars) obtained in the choice phase and pseudo-choice rates (filled circles) derived from the estimated values (Eqs. 3 and 4). **B, D** Correlation between the distractor effect (Eq. 2) and "*HV − LV*" estimate difference between two contexts. Error bars in (**A**) and (**C**) indicate standard errors of the mean across participants and in (**B**) and (**D**) the blue dots indicate individual participants.

### Common value representations in estimation and choice

To recap the behavioral findings, the estimation data revealed a consistent contextual modulation, with high distractor values leading to an overall underestimation of the two high-value alternatives. There was also a weak overall reduction in the estimated value difference of the two high-value alternatives, although this was present only in Experiment 2. By contrast, the choice data did not reveal statistically significant evidence for consistent contextual modulation (2-way ANOVA on $p(HV\ over\ LV)$, aggregated across the two experiments: context: $F(1, 388) = 0.02$, $p = 0.89$, $\eta^2 < 0.01$, 95% CI = [0.00, 0.008], trial-type: $F(1, 388) = 1.07$, $p = 0.30$, $\eta^2 < 0.01$, 95% CI = [0.00, 0.02], interaction: $F(1, 388) = 0.11$, $p = 0.74$, $\eta^2 < 0.01$, 95% CI = [0.00, 0.01], and Bayesian ANOVA with $P(M) = 0.2$ for all models on $p(HV\ over\ LV)$: null model: $BF_{01} = 1.00$, context model: $BF_{01} = 8.86$, trial-type model: $BF_{01} = 5.32$, full model with interaction: $BF_{01} = 295.21$). Should behavioral patterns in the estimation and choice phases be interpreted within a single explanatory framework? Addressing this question requires determining whether behavioral outputs in estimation and choice were governed by common value representations.

To determine that, we examined the consistency between value estimates and choices. We applied an established approach (Eqs. 3 and 4 [30]) to calculate pseudo-choices that would have been made had participants relied on the values reported in the estimation phases. We then compared the choice rate between the actual choices and the pseudo-choices. Assessing the choice rate allowed us to combine binary and ternary choices in one quantity. Figure 4A, C illustrates that pseudo-choices obtained from the value estimates are highly correlated with the actual choices in both experiments (Pearson correlation, in Experiment 1 with First estimation: $r = 0.994$ (95% CI = [0.991, 0.996]), two-sided $t$-test on correlation coefficients: $t(29) = 595.62$, $p < 0.001$, $d = 108.75$, 95% CI = [108.37, 109.12], with Second estimation: $r = 0.992$ (95% CI = [0.987, 0.998]), $t(29) = 363.69$, $p < 0.001$, $d = 66.40$, 95% CI = [66.03, 66.77]; and in Experiment 2 with First estimation: $r = 0.96$ (95% CI = [0.93, 0.98]), $t(67) = 66.75$, $p < 0.001$,

$d = 8.09$, 95% CI = [7.85, 8.34], with Second estimation: $r = 0.93$ (95% CI = [0.89, 0.98]), $t(67) = 37.94$, $p < 0.001$, $d = 4.60$, 95% CI = [4.36, 4.84]). We further correlated the "*HV − LV*" difference between the two contexts from the first estimation phase against the distractor effect (Eq. 2) quantified in the choice phase, after combining binary and ternary choice trials (Fig. 4B, D, Pearson correlation between the two quantities in Experiment 1 using the First estimation data: $r_{(30)} = 0.57$, $p < 0.01$, 95% CI = [0.26, 0.77], and the Second estimation data: $r_{(30)} = 0.65$, $p < 0.001$, 95% CI = [0.38, 0.82]; and in Experiment 2 the First estimation data: $r_{(68)} = 0.4$, $p < 0.001$, 95% CI = [0.18, 0.59], and the Second estimation data: $r_{(68)} = 0.32$, $p < 0.01$, 95% CI = [0.09, 0.52]). This significant correlation, in conjunction with the pseudo-choices analysis above, indicates a non-negligible link between the value estimates and choices. These results motivate us to develop a single model in the next section that can jointly explain the behavioral patterns obtained in the estimation and choice phases. We note in passing that, despite the significant correlations between the estimation and choice distractor effects—which establish that the value representation is common across the two tasks—choices may introduce small systematic shifts in the direction of the effect. Such shifts could be induced via reinforcement learning mechanisms [32] owing to the dwarfed value representations in the high-distractor context, which is known to affect choice accuracy [45,46]. Choice-induced effects of this type could perhaps explain why the marginal negative distractor effect in estimation disappears in choices in Experiment 2. However, assessing these subtler patterns is beyond the scope of this paper.

### Context-dependence beyond normalization theories

Taken together, across two experiments, the value estimates of the two high-value alternatives were consistently reduced as the value of the distractor alternative increased (Fig. 3A, D). Regarding the distractor effect (i.e., the difference in value estimates or, equivalently, the relative choice share as a function of distractor value) there were considerable individual differences in the direction of the effect (Fig. 4B, D).

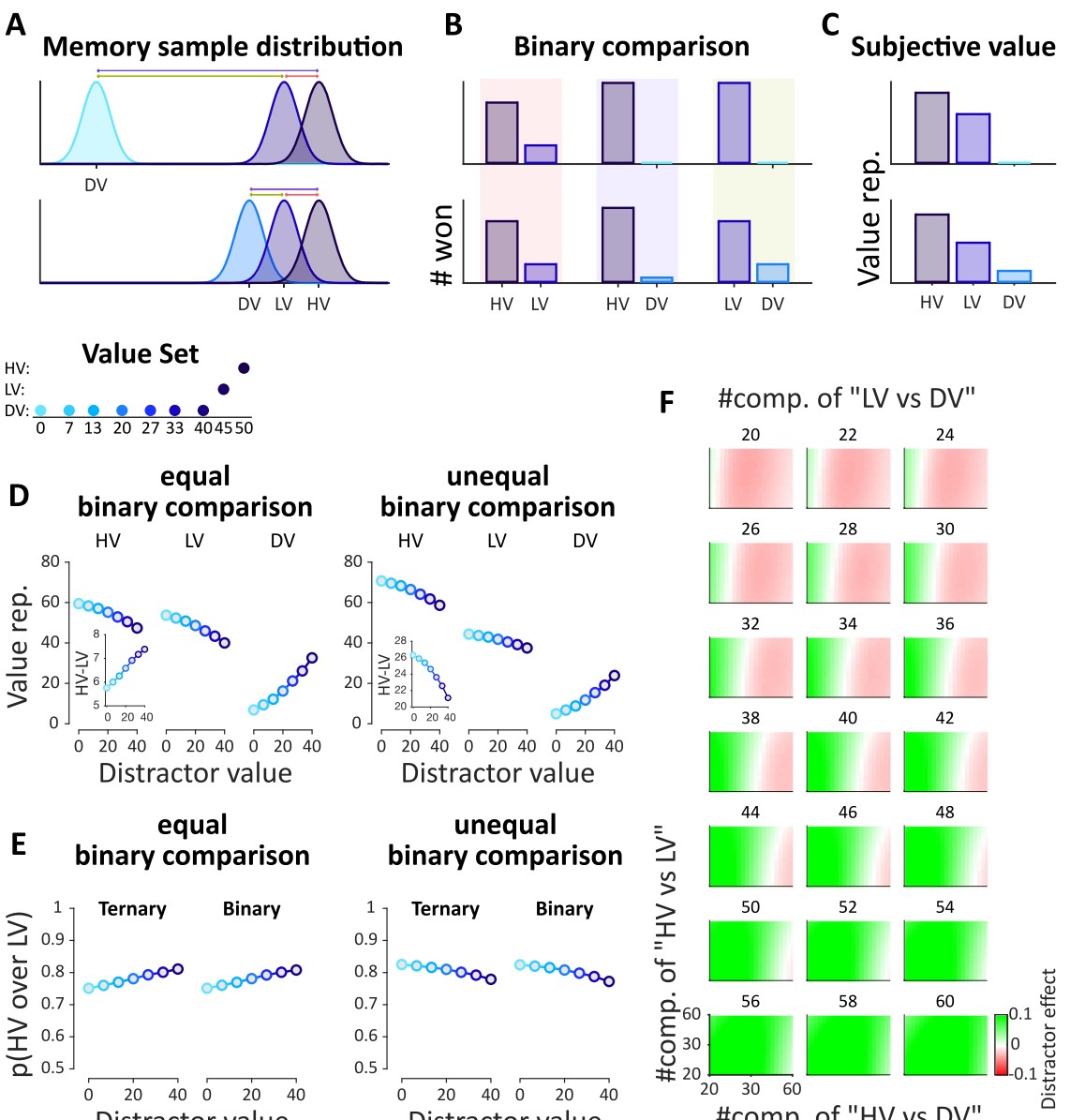

**Fig. 5 | Rank-based (RB) model illustration. A** The distribution of memory samples of the objective values. The top and bottom rows are in the presence of a low- and high-value distractor respectively. **B** The schematic of the number of times an alternative won in arbitrary 100 binary comparisons ($k$ parameter in the RB model), here all possible binary comparisons were presented. For example, in the bottom row, the HV once compared with LV (indicated by red arrow/shade) and once with DV (indicated by purple arrow/shade). **C** The final value representation of each alternative. Here for example the value of HV is the sum of the number of times HV won against LV and DV. **D** Value representations for each alternative in RB when the number of binary comparisons ($k$ parameter in the RB model) is equal for all pairs in left and when it is unequal, here $k_{HVvsLV} > k_{HVvsDV} > k_{LVvsDV}$. Like Fig. 1, the high-value (HV) and low-value (LV) alternatives were kept fixed and the distractor value

(DV) changed. In both conditions, as the distractor value increases the RB model predicts a decrease of HV and LV values. The embedded plots depict that the difference between the value representation of HV and LV increases in the equal condition and decreases in the unequal condition. **E** Simulated relative choice of HV over LV in the "temporal-context" framework. In each subplot, *P(HV over LV)* is calculated separately for ternary (HV, LV, DV) and binary choice trials (HV, LV). **F** Simulating distractor effect as the function of the number of binary comparisons for each pair, in each panel y-axis is the number of HV vs LV comparisons, the x-axis is the number of HV vs DV comparisons and the title is the number of LV vs DV comparison (for further details, see "Model simulations" section and Fig. S2 in Supplemental Information).

Divisive normalization can account for the overall value reduction and for participants who exhibited a negative distractor effect, but it fails to capture those with a positive distractor effect. Range normalization cannot capture the overall value reduction effect and can only capture participants with a positive distractor effect. A successful model should be able to produce an overall value reduction and, depending on parameter values, account for both positive and negative distractor effects.

We propose a rank-based (RB) model[34] that can capture these patterns in both estimation and choice data. This model relies on the decision-by-

sampling framework that has been successfully used to predict context effects in various settings, including multi-alternative and multiattribute choices[13,34,47]. The rank-based (RB) model assumes that the value representation of an alternative following the learning phase is constructed via a series of stochastic binary comparisons between memory samples of the target and of previously encountered alternatives (Fig. 5A, B). The constructed value of a target alternative is simply the number of times it won in these binary comparisons (Fig. 5C). The relative reduction of higher-value (HV) and lower-value (LV) changes as a function of the number of binary

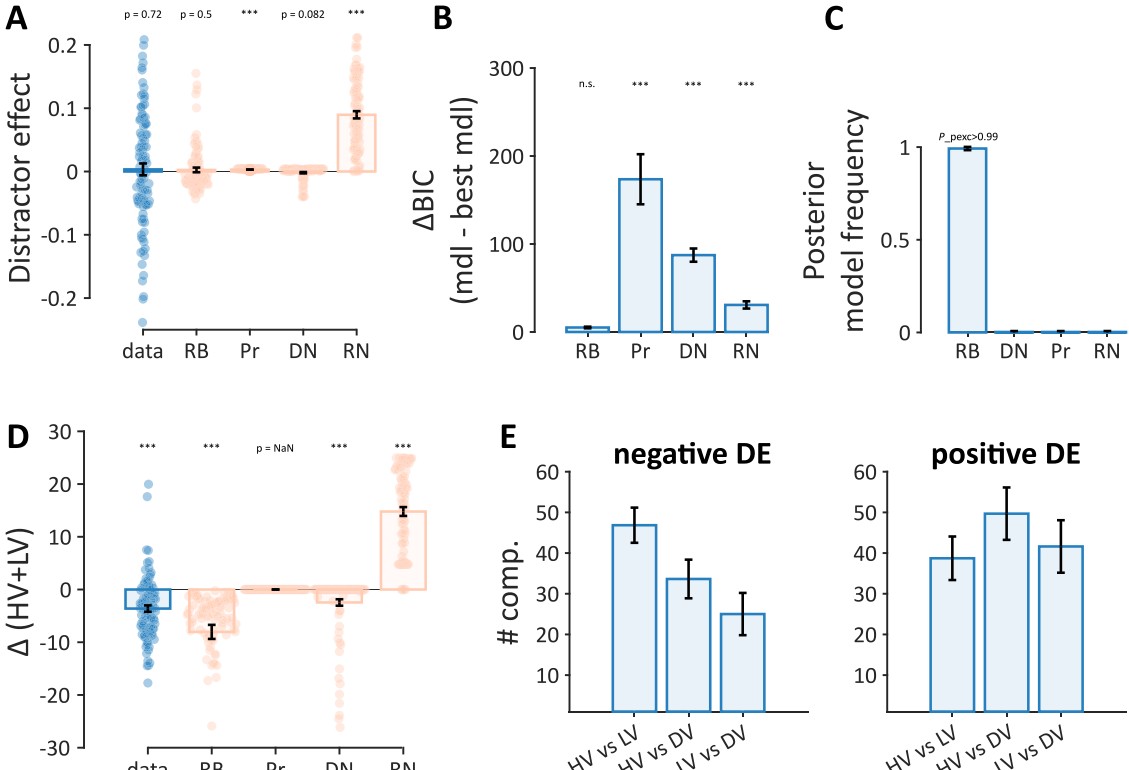

**Fig. 6 | Model comparison. A** Distractor effect comparison between the actual data and simulated data using the best-fitted parameters, the RB model has a closer fit to the actual distractor effect in the choice data. **B** Fixed-effect Bayesian model comparison, Black asterisks: significant ΔBIC against 10, one-sided one-sample *t*-tests[48]. **C** Random-effect Bayesian model comparison[43]. **D** The difference of "*HV* + *LV*" between two contexts, simulated by the best-fitted parameters of the RB model.

**E** Number of binary comparisons (best-fitted *k* parameters in RB model) for two groups of participants with negative and positive distractor effects. In all panels, the data here is pooled across both experiments (*N* = 98) (Fig. S3 in Supplemental Information for each Experiment), the error bars are standard errors of the mean across participants, and the colored dots indicate the individual participant.

comparisons made between each of the three possible pairs. Figure 5D shows that a balanced number of binary comparisons across all pairs increases "*HV* − *LV*" (Fig. 5D, left, embedded plot) when the distractor is higher, giving rise to a positive distractor effect. This is because the lower-value (LV) alternative, being closer to the distractor value, receives stronger competition from the distractor value than the higher-value (HV) does. This larger penalty the lower-value (LV) pays can be reversed when the *LV* vs. *DV* binary comparison is the least frequent one. In this unbalanced case "*HV* − *LV*" decreases (Fig. 5D, right, embedded plot) when the distractor is higher, giving rise to a negative distractor effect. In both the balanced and unbalanced cases, the constructed values are overall reduced for both high-value alternatives (HV and LV) due to increased competition with the distractor value (DV) in the high-distractor context (Fig. S2 in Supplemental Information).

We fitted the rank-based (RB) model and the other models depicted in Fig. 1 to the choice data from both experiments. As shown in Figs. 1 and 5, each of the models gives rise to different value representations as a function of the distractor value. Given the lack of immediate-context effects in the choice phase, these value representations are assumed to have taken shape by the end of the learning phase; and are probabilistically translated onto choices via a context-independent choice function (i.e., a probit, Eq. 5) in both binary and ternary trials. Figure 6A depicts that the rank-based (RB) model provided a closer approximation to the choice distractor effect pooled across both experiments (Fig. S3 in Supplemental Information for each Experiment) relative to the other models; we show this visually by simulating choices using the best-fitted parameters of each model, and then pooling the model predictions across both ternary and binary choices given that there was no difference in the distractor effect between ternary and binary choices. In particular, the rank-based (RB) model is the only model that captures both negative and positive distractor effects. Two further

model comparisons confirmed formally that the rank-based (RB) model accounted for the choice data better than the alternative models (Fig. 6B, C). First, in a fixed-effects comparison, the rank-based (RB) model has the smallest difference relative to the best-individualized model (ΔBIC > 10[48]) across both experiments (RB, Pr, and RN had the smallest BIC for 53.3%, 2%, and 26.7% of the participants in Experiment 1, and for 64.7%, 0.03%, and 32.3% of the participants in Experiment 2). Second, in a random-effects model comparison, the rank-based (RB) model is better supported (*P*pexc > 0.99[44]) in both experiments. Note that the rank-based (RB) model can describe the relative choice of higher-value over lower-value (Eq. 1) well in ternary and binary trials as we expected (two-sided paired *t*-test on predicted *p*(HV over LV) in Ternary: *t*(97) = −0.87, *p* = 0.39, *d* = 0.09, 95% CI = [−0.11, 0.29], and Binary: *t*(97) = −0.22, *p* = 0.83, *d* = 0.02, 95% CI = [−0.18, 0.22], Fig. S3 in Supplemental Information for each Experiment).

Figure 6D shows that the rank-based (RB) model predicts a large decrease in the estimated values of the two high-value alternatives (HV and LV) as the distractor value increases (DV), equivalent to what we observed in the behavioral data (Fig. 3A). In Fig. 6E, we demonstrate how rank-based (RB) can capture the positive and negative distractor effect as a function of the distribution of binary comparisons across the different pairs. Here we looked at the fitted parameters *k* (Eq. 9) for participants showing negative (with DE lower than the median) and positive (with DE lower than the median) distractor effects, in the left and right panels respectively. Consistent with the qualitative predictions of the model depicted in Fig. 5, Fig. 6E shows a significant imbalance in the distribution of binary comparisons (1-way ANOVA: *F*(2, 144) = 5.32, *p* < 0.01, partial *η*² = 0.07, 95% CI [0.01, 0.09]) for the group of participants with negative distractor effect; and a balanced distribution (1-way ANOVA: *F*(2, 144) = 0.87, *p* = 0.42, partial *η*² = 0.01, 95% CI [0.00, 0.04]) for the group of participant with positive distractor effect (Table 1 and Fig. S4 in Supplemental Information for the fits

**Table 1 | Best-fitted model parameters, (mean ± standard error of the mean across participants, aggregated across both experiments, *N* = 98)**

| Model Parameter | RB | Pr | DN | RN |
|---|---|---|---|---|
| $k$ | - | - | 92.34 ± 5.08 | - |
| $k_{HVvsLV}$ | 42.78 ± 3.45 | - | - | - |
| $k_{HVvsDV}$ | 41.66 ± 4.06 | - | - | - |
| $k_{LVvsDV}$ | 33.32 ± 4.20 | - | - | - |
| $\tau$ | 54.48 ± 4.13 | - | - | - |
| $\sigma$ | - | - | 87.59 ± 5.78 | - |
| $w$ | - | - | 0.18 ± 0.04 | 0.28 ± 0.04 |
| $\sigma^2_f$ | 79.77 ± 2.18 | 60.67 ± 3.66 | 55.72 ± 2.59 | 20.64 ± 3.15 |

of the other models and the best-fitted parameters of all models). Thus, the rank-based (RB) model can reconcile an overall value reduction with negative or positive distractor effects in estimation (and consequently in choice, Fig. 6F) under a single mechanism (Fig. S2).

## Discussion

Previous findings suggest that the value of an inferior (distractor) alternative influences choices between two high-value alternatives[3,6,49]. However, the direction of the distractor influence remains generally contentious while its timescale in value-learning tasks is underexplored[16,28,29]. Using a value-learning paradigm that was previously shown to induce a negative distractor effect in primates[6], we revealed an average null effect with considerable individual variability in its direction in human choices. We observed equivalent variability when quantifying the distractor effect using the value estimates participants provided immediately after the learning phase. The estimation data further revealed a consistent distortion, characterized by an overall reduction of value for the two high-value alternatives in high-distractor contexts. Further, we found that choice behavior was indistinguishable between binary (distractor absent) and ternary trials (distractor present), indicating that changes in the immediate choice context have no impact on behavior. Taken together, these findings suggest that the influence of the distractor alternative is situated in the broader temporal context, and varies in direction across individuals but remains consistent within individuals.

Context effects have been widely reported in psychology and behavioral economics in various domains including multiattribute consumer choices[50,51] and risky choices[3,8,21,52]. Typically, these violations are obtained in single-shot decisions, where participants need to compute the value of novel alternatives (based on subjective or externally determined criteria) on a single trial basis. Distractor effects, which is a certain type of context effects, have been reported both in single-shot decisions[25,30,32] and in decisions that tap upon learned values. Finding that in our learning task, the immediate context does not exert any influence on choices does not automatically overrule previous distractor effects obtained in single-shot tasks. It rather suggests that in value-learning tasks, where choices are guided by past rewards, the relative influence of the immediate context dissipates. This null immediate-context effect could mean that the entire set of alternatives could be kept in memory and guide value computations, even in binary trials. Value-learning tasks that tax the memory by involving a larger number of alternatives could perhaps reveal a more complex interplay between the influence of the temporal and the immediate context. Additionally, our study displayed rewards using perceptual stimuli (i.e., moving dots) in order to emulate the sensory uncertainty associated with the consumption of a primary reinforcer in Louie et al.[6]. An open empirical question is whether displaying rewards using stochastically sampled numerical values (a common practice in human value-learning tasks[30,53]) would produce different behavioral patterns.

Although the design of our study closely followed the design in Louie et al.[6], we did not replicate the strong negative distractor effect they found in the choice behavior of two monkeys. The negative distractor effect in Louie et al. was described as an immediate-context effect attributed to divisive normalization. However, recent research[32] has shown that a negative distractor effect can emerge without the recruitment of normalization mechanisms. This can occur when learning is conducted through choices among two or more alternatives, with reward feedback delivered exclusively for the chosen alternative. In these cases, in a high-distractor context, the distractor alternative is chosen (sampled) more frequently compared to a low-distractor context. As a consequence, participants in the high-distractor context sample the two high-value alternatives less often, exhibiting a lower choice accuracy due to increased value uncertainty (i.e., an "emergent" negative distractor effect). While in Louie et al. single-item learning trials preceded the choice phase, it is conceivable that the two monkeys continued to learn the values of the alternatives from the partial feedback received in the choice phase. This could have led to an "emergent" negative distractor effect owing to compromised learning in the high-distractor context[32]. The rapidly stabilized learning in our task prevented an "emergent" effect from arising. Thus, the discrepancy between our findings and those in Louie et al.[6] can be reconciled if the negative distractor effect was "emergent" in their case.

A central finding in our study was that value estimates and choices did not align with either classic positive or negative distractor effects. Instead, we observed considerable individual variability, with different participants exhibiting either positive or negative effects. Individual variability has also been recently reported in multiattribute context effects[54,55], overturning a long-held assertion about the directional nature of certain context effects. Thus, our findings, together with those from the multiattribute literature, begin to paint a richer empirical landscape. Future work could shift away from theoretical frameworks making rigid, one-directional predictions and instead focus on understanding the neural and computational mechanisms that give rise to variability in context-dependent valuation. Across these lines, inspired by the decision-by-sampling and "frequency" models[33,34,56], we developed a stochastic rank-based model that encompasses both positive and negative distractor effects while predicting a robust value reduction in high-distractor contexts, similar to the behavioral data.

## Limitations

While this study provides insights into the influence of an inferior alternative on value learning and decision-making, it also has specific limitations. First, as our aim was to semi-replicate the design of the monkey study[6], our experimental design, including only three alternatives per mini-block, may have been relatively simple for human participants, possibly limiting its generalizability. Second, time constraints during the pandemic required us to run the task in a single session (~3 h), limiting the number of difficulty levels and trials we could include. Future work should explore tasks with a larger range of difficulty levels to assess the robustness of the observed effects. Third, future research could extend our findings by considering different types of rewards, both deterministically and probabilistically delivered, as this could offer insights into how various reward structures influence value learning and decision-making. Lastly, although the model we proposed here was descriptively adequate across two experiments, future work could test its generalizability to existing datasets and, by extension, its validity as a general-purpose value-learning model.

## Conclusion

To conclude, our findings suggest that during value-learning, the value of an inferior (distractor) alternative distorts value representations over a long timescale spanning several learning and choice trials. Crucially, the direction of this distortion is subject to across-participants variability, which divisive and range normalization theories cannot capture. Instead, this variability is best captured by a mechanism that constructs subjective value representations by comparing the value of the target alternative with the value of other alternatives, sampled from the memory context. These findings shed new

light on the mechanisms that govern value learning and decision-making among multiple alternatives.

## Data availability
Primary data of all participants supporting the findings of this study are available on OSF.

## Code availability
All analysis scripts supporting the findings of this study are available on OSF.

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

## Acknowledgements

This work was supported by the EU Horizon 2020 Research and Innovation Program (ERC starting grant no. 802905) to K.T. The funders had no role in the study design, data collection and analysis, decision to publish, or preparation of the manuscript. We thank Yinan Cao for generously providing us with his insights during the early stage of this research.

## Author contributions

Maryam Tohidi-Moghaddam: Conceptualization, Methodology, Software, Validation, Formal Analysis, Investigation, Data Curation, Writing – Original Draft, Writing – Review & Editing, Visualization. Konstantinos Tsetsos: Conceptualization, Methodology, Validation, Writing – Original Draft, Writing – Review & Editing, Supervision, Project Administration, Funding Acquisition.

## Funding

## Competing interests

The authors declare no competing interests.
