## [Transparent Peer Review file · Communications Psychology]

The timescale and direction of influence of a third inferior alternative in human value-learning

Corresponding Author: Ms Maryam Tohidi-Moghaddam

Version 0:

Decision Letter:

Dear Ms Tohidi-Moghaddam,

Thank you for your patience during the peer-review process. Your manuscript titled "The timescale and functional form of context-dependence during human value-learning" has now been seen by 3 reviewers, whose comments are appended below. You will see that they find your work of some potential interest. However, they have raised quite substantial concerns that must be addressed. In light of these comments, we cannot accept the manuscript for publication, but would be interested in considering a revised version that fully addresses these serious concerns.

We hope you will find the Reviewers' comments useful as you decide how to proceed. Should additional work allow you to address these criticisms, we would be happy to look at a substantially revised manuscript. If you choose to take up this option, please highlight all changes in the manuscript text file, and provide a detailed point-by-point reply to the reviewers.

Editorially, we consider the following concerns needs to be carefully addressed:

- 1) All reviewers agree the new model proposed in the manuscript has potential, but that data presented in the manuscript is not compelling evidence. We ask that in revision, you present new empirical data from a well-powered replication study to provide stronger evidence for the model (and effect). The replication study needs to be powered at $\geq 80\%$ to detect the target effect size. The target effect size should be the smallest effect size of theoretical interest, not effect size established in the present work. Note that in the absence of a justified diversion from the standard alpha level, only results that are significant at $p < 0.05$ (in NHST) may be interpreted.
- 2) All reviewers point out the behavioral effect representing the behavioral signature predicted by the model appears to be rather weak and Reviewer#3 suggests that this may be due by the nature of the reward signal, an issue also mentioned by Reviewer #2. The replication study could directly test the impact of this choice by varying typical reward feedback across conditions (note that this design change will affect the sample size).
- 3) Finally, we ask that you respond to the referees' requests for additional analyses (where these rule out potential alternative explanations) and better integration of existing literature.

I am attaching a checklist that details critical reporting requirements for the revised manuscript. Please attend to each item and ensure your manuscript is fully compliant. We are requesting that your manuscript aligns with these requirements as this facilitates the evaluation of your manuscript, reducing delays in re-review and potential future acceptance. If your revised manuscript is not aligned with these requests on major issues, such as those concerning statistics, it may be returned to you for further revisions without re-review. Additional information can be found in our style and formatting guide [Communcations Psychology formatting guide](https://www.nature.com/documents/commspsychol-style-formatting-guide-accept.pdf).

If the revision process takes significantly longer than five months, we will be happy to reconsider your paper at a later date, provided it still presents a significant contribution to the literature at that stage.

Please use the following link to submit your

- revised manuscript,
- point-by-point response to the referees' comments,
- cover letter (as a separate document),
- the Editorial Policy Checklist (see below),
- the Reporting Summary (see below), and
- the completed Editorial Request Table (attached):

Link Redacted

Thank you for the opportunity to review your work.

Best regards,

Eva R. Pool

Eva R. Pool, PhD
Editorial Board Member
Communications Psychology
orcid.org/0000-0001-5929-1007

REVIEWER EXPERTISE:

Reviewer #1, Reviewer #2, Reviewer #3: Decision making and reinforcement learning

REVIEWER REPORTS:

Reviewer #1 (Remarks to the Author):

In the present manuscript, the authors investigate the origin of distractor effects in learning tasks. Specifically, they test the hypothesis that the temporal context influences value estimation and comparison, rather than or in addition to the immediate choice context. They compare a range of candidate models to explain any distractor effects. Their empirical results are inconsistent with any of these previous models, so the authors derive a new model inspired by decision by sampling that can explain the observed pattern: a large devaluation for larger distractor values and a small reduction in the value difference. Their model can explain the absence of the latter in value estimation but not choice under the assumption that value estimation is subject to greater response noise.

This is a compelling paper, albeit with somewhat limited scope given the target journal. For a purely behavioral study, a single experiment with 30 participants seems fairly limited, particularly given that the authors find the opposite of the original finding in monkeys and given that the model they propose is post-hoc. It also leaves open many questions as to how the distractor effects observed in other settings come to be and whether the proposed mechanism adds to other mechanisms in these settings.

I am also a bit concerned about some aspects of the task that may nudge the current findings in this direction, specifically the perceptual nature of the reward feedback. Why were noisy random dot stimuli chosen for the reward feedback?

To validate the claim that estimation may be associated with greater response noise, can the authors show that a) when simulating such an estimation process the effect does indeed go away and b) in an additional experiment that reduces response noise (e.g. by reducing the number of response options on the scale - which I understand is currently continuous) that they can show this difference effect during estimation as well?

I find the paper really quite intriguing but I feel it leaves too many questions unanswered and given the novelty of the effect it would be good to show that the effect is robust and the model validated in a replication.

Reviewer #2 (Remarks to the Author):

This study investigated the context-dependence of value coding in decision-making and proposed a new model concerning context-dependent valuation and choice. The findings shed light on how distractors influence seemingly irrelevant choice options, presenting an intriguing perspective. While the model comparison was conducted rigorously, ensuring the validity of the model selection, there are critical issues that require attention.

1. The positive effect of the distractor stands out as a key behavioral observation utilized in discerning the model. However, as indicated by the authors, this effect appears a trend-level. It is crucial for this effect to be robust or distinct from predictions made by other models, criteria that currently do not seem to be met based on the behavioral outcomes and Figure 6A. Given that the simulations depicted in Figure 6A cannot be altered, conducting an additional experiment, such as increasing the sample size or replicating the study as an independent experiment, becomes imperative.
2. The sample size of 30 appears rather small. Justification for this diminutive sample size should be provided based on power calculation. If such justification is lacking, collecting additional data becomes imperative. Conducting a power calculation prior to any further data collection is essential to ascertain an adequate sample size, ensuring sufficient statistical power for meaningful analyses, which would help the issue #1 hopefully.
3. The simulation depicted in Figure 1 does not include the simulation of the RB model. Including a systematic simulation of the RB model would significantly enhance the interpretability of the results, allowing readers to grasp the distinctions among the different models more clearly. Moreover, it could apply for other models, as those model have not been simulated with the actual values that were used in the experiment. By providing these additional results, readers would gain a deeper understanding of the comparative performance of the models.
4. In the Method and Participants section, I'm curious whether the data was collected concurrently with fMRI data. If this was the case, it would be prudent to explicitly mention it in the manuscript and elaborate on how the other part of the data was managed.
5. The title appears confusing as the paper and the model focus on the choice behavior. Additionally, the term "functional form" is not clearly defined, leaving readers uncertain about its meaning in the context of the study.

Minor points

1. In figure 2D, the last part of choice trial should have two screens for "feedback" and "no feedback" conditions

Reviewer #3 (Remarks to the Author):

This paper investigates context-dependent learning and choice behavior, demonstrating that subjective value estimates are distorted by inferior (distractor) options. The data suggest that context effects emerge early in the task, during the initial learning phase, and that the immediate context (i.e., set of available options) has little to no effect, at least when temporal context is also available. The authors show that their data conflict with the predictions of both divisive and range normalization, and are instead better explained by a memory-based model based on stochastic binary comparisons.

Overall, I thought the topic of the paper was very interesting and relevant to current debates in value-based decision research. The paper is full of nice figures that make the results and models clear and easy to follow. The statistical analyses appear to be appropriately conducted. Below are some comments and suggestions, divided into major/minor, that I hope will be helpful:

Major comments:

1. The authors describe the memory-based model as new; however, a very similar model was recently proposed in Hayes and Wedell (2023). Their "Frequency model" assumes that outcomes observed on each trial are compared to a sample of previous outcomes from the local outcome distribution, drawn from memory. These comparisons are assumed to be binary, in the tradition of decision by sampling and range-frequency theory. As a result, options are subjectively valued based, in part, on how their outcomes rank within the local outcome distribution. Although the Frequency model is not mathematically identical to the model in the present paper, it is similar enough that it should be acknowledged.

References:

Hayes, W. M., & Wedell, D. H. (2023). Testing models of context-dependent outcome encoding in reinforcement learning. *Cognition*, 230, 105280.

See also Hayes, W. M., & Wedell, D. H. (2023). Effects of blocked versus interleaved training on relative value learning. *Psychonomic Bulletin & Review*, 30, 1895-1907.

2. On p. 4, the authors state that "previous studies have not systematically examined the timescale at which context-based distortions operate." Then, a bit later, they state that "the distinction between temporal and immediate context is blurry" in tasks that involve learning the values of options within separate temporal contexts (e.g., Bavard & Palminteri, 2023; Hayes & Wedell, 2023, etc.). I would argue that those studies have demonstrated a strong effect of temporal context— e.g., participants often end up "irrationally" preferring a lower value option over a higher value option simply because the former was originally paired with an even worse option, and the latter with an even better option, during the initial learning phase. This kind of effect can only be attributable to temporal context.

3. The evidence for a context effect in choice was rather weak (anecdotal evidence for a positive distractor effect). Some of this could be due to being underpowered. However, I wonder if a larger context effect would be observed using numerical rewards (instead of dots; see next point) and complete feedback instead of partial feedback? Previous research has found that context effects in these types of tasks are generally more pronounced with complete feedback (Bavard et al., 2018; 2021). I think running an additional experiment with these parameters, if possible, might help to bolster the overall conclusions.

References:

Bavard, S., Lebreton, M., Khamassi, M., Coricelli, G., & Palminteri, S. (2018). Reference-point centering and range-adaptation enhance human reinforcement learning at the cost of irrational preferences. *Nature Communications*, 9, 4503.

Bavard, S., Rustichini, A., & Palminteri, S. (2021). Two sides of the same coin: Beneficial and detrimental consequences of range adaptation in human reinforcement learning. *Science Advances*, 7, eabe0340.

4. The rationale for using moving dots instead of numerical values as rewards should be explained, given that this diverges from most other studies in this area.

5. Can the authors discuss whether the memory process assumed by the rank-based model occurs at the feedback encoding stage or the choice stage? The paper seems to imply that it occurs at the time of choice; however, this would not be able to explain the kinds of irrational transfer preferences that have been observed in prior studies and described in point (2) above. That is, participants in those studies sometimes form a strong preference for A over B, even though all of A's rewards were smaller than B's rewards. The only way this could occur is if the subjective values of the rewards were distorted by the local context at encoding time.

Minor comments:

1. Shouldn't the scale of the value representations in Figure 1A (y-axis) be between 0 and 1 for divisive and range normalization?
2. Is Figure 2C how the estimation trials actually appeared? I was confused by the fact that in the figure, the scale goes from 4 dots, to 6, to 15 with equal spacing.
3. P. 8, line 170: "using a two-sided t-test on." This sentence may have been cut off.
4. Very minor, but I don't think the units (ms) need to be in parentheses when they appear in confidence intervals. Removing the parentheses may make it less crowded.
5. What should we make of the general tendency to overestimate the values of options (Fig. 3A)?

EDITORIAL POLICIES

We ask that you ensure your manuscript complies with our editorial policies and reporting requirements.

To that end, we require revised manuscripts to be accompanied by two completed items: a reporting summary that collects information on study design and procedure, and an editorial policy checklist that verifies compliance with all required editorial policies

- <https://www.nature.com/documents/nr-reporting-summary.zip>>Nature Research Reporting Summary
- <https://www.nature.com/documents/nr-editorial-policy-checklist.pdf>>Editorial Policy Checklist

All points on the policy checklist must be addressed. Your revised manuscript can only be sent back to the referees if these checklists are completed and uploaded with the revision.

Notes: If you have submitted a Stage 1 Registered Report, Review, Primer, Comment, or Perspective you do not need to submit these forms. If you have already submitted these forms, you may disregard this request.

Version 1:

Decision Letter:

Dear Ms Tohidi-Moghaddam,

Your manuscript titled "The timescale and direction of influence of a third inferior alternative in human value-learning" has now been seen by our reviewers, whose comments appear below.

You will see that the reviewers have no further requests. However, as highlighted below and in the attached checklist, there are a number of significant editorial concerns about the reporting of statistics and the interpretation of non-significant findings in null-hypothesis significance tests. Although we recognize that the computational model largely supports your interpretation of the results, we highlight that our in-principle acceptance of your study is conditional on the satisfactory address of all remaining concerns.

We therefore invite you to revise your paper one last time to address the remaining concerns. At the same time we ask that you edit your manuscript to comply with our format requirements and to maximise the accessibility and therefore the impact of your work.

EDITORIAL REQUESTS:

In the present version of the manuscript, there are frequent statements of the absence of an effect or difference on the basis of non-significant results derived from null-hypothesis significance tests (NHST). The journal's policies on statistics and reporting do not permit this.

In the case of ancillary tests that are not further interpreted and do not inform the key hypotheses, this means that you must at least revise the wording. Instead of stating that the ns finding in NHST provides evidence for conditions being equal or indistinguishable, you must highlight that there was no statistically significant evidence for a difference.

For key results, such as those summarized in the sections "No distractor effect in the choice phase" and "Common value representations in estimation and choice", you must provide positive evidence for the absence of a difference. You may use either Bayesian statistics (with a $BF_{01} > 3$ criterion) or equivalence tests. You must also provide a sensitivity analysis, specifying how you defined the smallest size of the effect of interest, what level of power your sample achieved for this target effect size, and based on what statistical model (please do not provide a post-hoc power analysis targeting the effect size that was achieved in the existing samples).

Please review our specific editorial comments and requests regarding your manuscript in the attached "Editorial Requests Table", which also contains more detail on the statistics issues. Please outline your response to each request in the right hand column. Please upload the completed table with your manuscript files as a Related Manuscript file.

SUBMISSION INFORMATION:

OPEN ACCESS:

Communications Psychology is a fully open access journal. Articles are made freely accessible on publication. For further information about article processing charges, open access funding, and advice and support from Nature Research, please visit <https://www.nature.com/commpsychol/open-access>

* **DATA AVAILABILITY:**

All Communications Psychology manuscripts must include a section titled "Data Availability" at the end of the Methods section. More information on this policy, is available in the Editorial Requests Table and at <http://www.nature.com/authors/policies/data/data-availability-statements-data-citations.pdf>

Link Redacted

Best regards,

Marika
on behalf of Eva Pool

Marika Schiffer, PhD
Chief Editor
Communications Psychology

Eva Pool, PhD
Editorial Board Member
Communications Psychology

REVIEWERS' COMMENTS:

Reviewer #1 (Remarks to the Author):

The authors have addressed my concerns in their revision. I believe the manuscript is stronger now and I am happy for it to be accepted in principle.
However, there seem to be empty panels in Figure 4 (B&D).

Reviewer #2 (Remarks to the Author):

Thank you for your effort in collecting new data and improving the quality of the paper. The modified version and additional study provide stronger evidence supporting the findings and makes the results more convincing.

Minor point:

I cannot see the dots in Figure 4b and 4d. I'm not sure whether this is a technical issue or a mistake.

Reviewer #3 (Remarks to the Author):

I appreciate the responses from the authors to my comments on the original version. I think the additions and clarifications in the revised version have significantly improved the manuscript, and I now recommend acceptance.

COMMSPSYCHOL-24-0106-T: Response to the reviewers' comments

Dear reviewers,

We thank you for the constructive evaluation of our work. We are delighted that you found our work interesting, and we appreciate the number of helpful suggestions you made for improvement. In the revised version, we have carefully addressed each of their points, which we believe has considerably strengthened our paper. In what follows, we first provide an overview of major revisions, followed by a detailed point-by point reply to each of your comments (printed in **blue bold italics**). We hope that you will now find our manuscript suitable for publication in Communication Psychology.

Sincerely,

Maryam Tohidi-Moghaddam, Konstantinos Tsetsos

Overview of major revisions

Following your recommendation, we conducted a new well-powered replication study, which allowed us to better gauge the robustness of the behavioral effects and the suitability of our proposed model. In sum, we collected new data from $N = 68$ participants based on a power analysis, powered at 90% for an effect size equal to 0.4. The experiment was conducted online using the Prolific platform for recruiting of participants (<https://www.prolific.com>). Details on the procedure of this online study are given in the revised *Methods* section of the manuscript and supplemental material.

Replicated findings

The statistically strongest findings of the onsite study (referred to as Experiment 1 in the revision) were replicated in the online study (hereafter, Experiment 2). First, we confirmed that the choice rates were indistinguishable between binary and ternary trials, which indicates that the immediate context (i.e., the availability of the distractor alternative for choice) had no impact. Second, we confirmed that the estimated values for the two high-valued options were lower in the high-distractor context. This replicated “estimation” effect is important as it challenges certain models and motivates the rank-based (RB) model we propose. Finally, we replicated the strong association between estimated values and choice rates.

Findings challenging our original interpretation

Findings of the onsite study pertaining to the distractor effect were not fully replicated. These effects refer to the estimated value difference between the highest (HV) and the second highest (LV) alternatives; and to the relative choice share of HV over LV. First, in experiment 2, we found that the estimation difference ($\Delta(\text{HV-LV})$) shrunk in the high-distractor context. This effect was weak but significant ($p = 0.03$), unlike in experiment 1 where the same effect was null ($p > 0.98$). In the revision, we briefly discuss potential reasons that might drive the marginal negative effect in Experiment 2 (e.g., overall lower accuracy in the online study). Second, in Experiment 2, we did not find a significant distractor effect on choices. This aligns with the non-significant distractor effect observed in Experiment 1. However, our original interpretation focused on the positive numerical value of this effect, which we interpreted as a weak “trend for a positive distractor” effect. As we describe in the next section, we now abolish this “trend for a positive distractor” interpretation.

Changes after considering the new experimental results

The new study prompted us to reconsider the previous statement about a weak “trend for a positive distractor” effect, and, more generally, to shift our perspective from focusing on interpreting the direction of the average distractor effect (in estimation and choice) to highlighting the individual variability associated with it. Indeed, when examining the variability among individual participants (Figures 4A, D), we see that some participants show a positive distractor effect while others show a negative distractor effect in both experiments.

Thus, the distractor effect is idiosyncratic across participants but is consistent within participants, across both estimation and choice tasks. In the revision, we adjusted the rank-based (RB) model so that it can extend to both positive and negative distractor effects, while still predicting an overall reduction in the estimated values in high-distractor contexts (Figure 5D and S2). The flexibility of the RB model stands in contrast with the rigid directional predictions extant normalization models make. We formally compared the updated RB model to existing normalization (divisive and range) models and reported strong support for the former.

Practically, the results of experiment 2 led to changes on the “functional form of context-dependence” aspect of our paper, which is a relatively small section. This is evident from the fact that only the last sentence of our Abstract needed to be revised. In terms of other sections, the Results section is now shorter, more focused, and better streamlined as we have removed the speculative discussion about the weak “trend for a positive distractor effect”. Certain aspects of the Discussion were revised accordingly.

We thank you once again for prompting us to conduct a replication study. This new study helped us broaden our interpretation of the empirical results and present a more confident conclusion focused on the individual variability associated with the distractor effect.

Point-by-point reply to reviewer #1

In the present manuscript, the authors investigate the origin of distractor effects in learning tasks. Specifically, they test the hypothesis that the temporal context influences value estimation and comparison, rather than or in addition to the immediate choice context. They compare a range of candidate models to explain any distractor effects. Their empirical results are inconsistent with any of these previous models, so the authors derive a new model inspired by decision by sampling that can explain the observed pattern: a large devaluation for larger distractor values and a small reduction in the value difference. Their model can explain the absence of the latter in value estimation but not choice under the assumption that value estimation is subject to greater response noise.

This is a compelling paper, albeit with somewhat limited scope given the target journal. For a purely behavioral study, a single experiment with 30 participants seems fairly limited, particularly given that the authors find the opposite of the original finding in monkeys and given that the model they propose is post-hoc. It also leaves open many questions as to how the distractor effects observed in other settings come to be and whether the proposed mechanism adds to other mechanisms in these settings.

Thank you for your positive evaluation of our paper and for the constructive suggestions. Now we have added a well-powered replication experiment whose results are largely consistent with our first experiment while also allowing us to draw more nuanced conclusion about the direction of the distractor effect. We describe these changes in “Overview of major revisions” section of this letter.

I am also a bit concerned about some aspects of the task that may nudge the current findings in this direction, specifically the perceptual nature of the reward feedback. Why were noisy random dot stimuli chosen for the reward feedback?

In this study, we aimed to replicate the monkey study conducted by Louie et al. (Louie et al., 2013) as similarly as possible for humans. Louie et al. delivered drops of juice to the monkey as reward. In the monkey study, in each block, the number of drops associated with an alternative was deterministic. However, the psychometric curves of the animals in this study suggests the presence of internal noise in the perception of rewards. Indeed, it is plausible to assume that at the sensory level, the same amount of reward might be registered differently from trial to trial (an assumption we informally corroborated in discussions with animal physiologists). To make the human reward feedback as similar as possible to the monkeys’ juice delivery, we needed to introduce some degree of “internal” uncertainty into the reward perception. Thus, we thought that instead of showing the dots statically on the screen for a fixed time (750 ms), we could present them in a dynamically moving format, in 10 consecutive frames of 75 ms each, to induce uncertainty at the perceptual level (We now clearly explain this informatio in the Methods of Supplemental material). Note that we did not use the classic noisy random dot motion used in classic decision studies (Gold & Shadlen, 2007). The reward size in our task was the number of dots, the more dots, the higher the value. The number of dots was kept fixed in each frame, and only the position of the dots changed from frame to

frame. We agree that an alternative, and perhaps more conventional, manipulation would be to draw the reward from a distribution, that way inducing external uncertainty. However, we were hesitant to use this manipulation because it would mark a significant departure from the initial monkey study, where rewards were deterministic and uncertainty was purely internal. We felt that an external uncertainty manipulation would introduce an additional layer of experimental complexity (i.e., controlling for sampling error, either by stretching the duration of the learning phase or by resorting to pseudorandom sampling). That said, we appreciate the reviewer's perspective and believe that repeating our experiment with an external uncertainty manipulation would be a very interesting future endeavour. Thus, we addressed this point on task parameters briefly in the Discussion.

To validate the claim that estimation may be associated with greater response noise, can the authors show that a) when simulating such an estimation process the effect does indeed go away and b) in an additional experiment that reduces response noise (e.g. by reducing the number of response options on the scale - which I understand is currently continuous) that they can show this difference effect during estimation as well?

We thank the reviewer for this very thoughtful comment and found the suggestions spot-on. However, considering the new replication data, we have removed our speculative discussion about the differences between estimation and noise. More broadly, we no longer try to reconcile discrepancies between the estimation and choice data. Instead, we emphasize the within-participants consistency between estimation and choice patterns. We acknowledge in the results section that additional processes may subtly cause patterns in estimation and choice to diverge. However, for the sake of brevity, we do not pursue this direction further.

I find the paper really quite intriguing but I feel it leaves too many questions unanswered and given the novelty of the effect it would be good to show that the effect is robust and the model validated in a replication.

We thank you for this invaluable suggestion, which we took on board. The replication study strengthened our interpretation of the empirical results and helped us refine the rank-based model we propose (please see "Overview of major changes" section).

Point-by-point reply to reviewer #2

This study investigated the context-dependence of value coding in decision-making and proposed a new model concerning context-dependent valuation and choice. The findings shed light on how distractors influence seemingly irrelevant choice options, presenting an intriguing perspective. While the model comparison was conducted rigorously, ensuring the validity of the model selection, there are critical issues that require attention.

Thank you for the thorough and constructive assessment of our work.

1. The positive effect of the distractor stands out as a key behavioral observation utilized in discerning the model. However, as indicated by the authors, this effect appears a trend-level. It is crucial for this effect to be robust or distinct from predictions made by other models, criteria that currently do not seem to be met based on the behavioral outcomes and Figure 6A. Given that the simulations depicted in Figure 6A cannot be altered, conducting an additional experiment, such as increasing the sample size or replicating the study as an independent experiment, becomes imperative.

2. The sample size of 30 appears rather small. Justification for this diminutive sample size should be provided based on power calculation. If such justification is lacking, collecting additional data becomes imperative. Conducting a power calculation prior to any further data collection is essential to ascertain an adequate sample size, ensuring sufficient statistical power for meaningful analyses, which would help the issue #1 hopefully.

We thank you for these two comments, which we took on board by conducting a larger replication study. You can find a summary of the results that this study yielded in the “Overview of major changes” section. To foreshadow, we no longer place undue emphasis on the trend-level positive distractor effect. Instead, we focus on the wide across-participant variability observed with respect to the distractor effect and adjust our rank-based model accordingly.

While we fully agree that a replication study was imperative given that some of our results were anecdotal, we respectfully disagree that our first study was underpowered. This is because each participant of the original study performed a large number of trials (c.f., the study lasted between 3 and 3.5 hours), which approximates experimental designs in psychophysics (Baker et al., 2021; Smith & Little, 2018). Our design resulted in each participant contributing highly precise behavioral metrics, that were subsequently submitted to group-level analyses. Conventional power analyses disregard the aspect of within-participant precision and focus exclusively on the number of individual participants.

3. The simulation depicted in Figure 1 does not include the simulation of the RB model. Including a systematic simulation of the RB model would significantly enhance the interpretability of the results, allowing readers to grasp the distinctions among the different models more clearly. Moreover, it could apply for other models, as those model have not been simulated with the actual values that were used in the experiment. By providing these

additional results, readers would gain a deeper understanding of the comparative performance of the models.

Thank you for raising these points. We agree that a figure illustrating the predictions of the RB model is needed. Given the structure of the paper and the overall narrative, we couldn't find an efficient way to add the RB predictions in Figure 1. Therefore, we show the RB predictions in Figure 5, as we introduce the RB model in the main text.

Regarding the simulation of the models, we would like to clarify that we based on the reward values we used in the experiment. The actual values in the experiment were as follows:

HV = {55, 50}, LV = HV-5 and DV = {18, 40}. For simplicity, in the simulation we used HV=50 and LV = 45, and varied the distractor value across a wider range {0, 7, 13, 20, 27, 33, 40} to provide a smoother illustration of the distractor effects.

4. In the Method and Participants section, I'm curious whether the data was collected concurrently with fMRI data. If this was the case, it would be prudent to explicitly mention it in the manuscript and elaborate on how the other part of the data was managed.

Thank you for this comment. We would like to clarify that this was a purely behavioral study.

5. The title appears confusing as the paper and the model focus on the choice behavior. Additionally, the term "functional form" is not clearly defined, leaving readers uncertain about its meaning in the context of the study.

Thank you for pointing this out. We have changed the title to "The timescale and direction of influence of a third inferior alternative in human value-learning". We found the new title more descriptive, clear and accurate.

Minor points

1. In figure 2D, the last part of choice trial should have two screens for "feedback" and "no feedback" conditions

Thanks for pointing this out, this is a great catch. We thought that adding a screen might visually overload the figure. Instead, we added text to the figure saying that the reward screen is skipped in No-Feedback mini-blocks. We paste the new figure here for your convenience:

Point-by-point reply to reviewer #3

This paper investigates context-dependent learning and choice behavior, demonstrating that subjective value estimates are distorted by inferior (distractor) options. The data suggest that context effects emerge early in the task, during the initial learning phase, and that the immediate context (i.e., set of available options) has little to no effect, at least when temporal context is also available. The authors show that their data conflict with the predictions of both divisive and range normalization, and are instead better explained by a memory-based model based on stochastic binary comparisons

Overall, I thought the topic of the paper was very interesting and relevant to current debates in value-based decision research. The paper is full of nice figures that make the results and models clear and easy to follow. The statistical analyses appear to be appropriately conducted. Below are some comments and suggestions, divided into major/minor, that I hope will be helpful:

Thank you very the positive assessment of our paper and for your encouraging words.

Major comments:

1. The authors describe the memory-based model as new; however, a very similar model was recently proposed in Hayes and Wedell (2023). Their “Frequency model” assumes that outcomes observed on each trial are compared to a sample of previous outcomes from the local outcome distribution, drawn from memory. These comparisons are assumed to be binary, in the tradition of decision by sampling and range-frequency theory. As a result, options are subjectively valued based, in part, on how their outcomes rank within the local outcome distribution. Although the Frequency model is not mathematically identical to the model in the present paper, it is similar enough that it should be acknowledged.

References:

*Hayes, W. M., & Wedell, D. H. (2023). Testing models of context-dependent outcome encoding in reinforcement learning. *Cognition*, 230, 105280.*

Thank you for pointing out this paper and model, which we unfortunately overlooked. We have now referenced this paper and explicitly mentioned the Frequency model in the discussion.

*See also Hayes, W. M., & Wedell, D. H. (2023). Effects of blocked versus interleaved training on relative value learning. *Psychonomic Bulletin & Review*, 30, 1895-1907.*

Thank you. This paper was referenced in the original paper in the introduction.

2. On p. 4, the authors state that “previous studies have not systematically examined the timescale at which context-based distortions operate.” Then, a bit later, they state that “the distinction between temporal and immediate context is blurry” in tasks that involve learning the values of options within separate temporal contexts (e.g., Bavard & Palminteri,

2023; Hayes & Wedell, 2023, etc.). I would argue that those studies have demonstrated a strong effect of temporal context— e.g., participants often end up “irrationally” preferring a lower value option over a higher value option simply because the former was originally paired with an even worse option, and the latter with an even better option, during the initial learning phase. This kind of effect can only be attributable to temporal context.

Thank you for pointing this out. Indeed, the way we phrased this part was ambiguous. What we intended to state was that in certain paradigms where learning and choices take place concurrently, the relative contribution of the temporal and immediate context is impossible to discern. We change the text to avoid the previous ambiguity. We now acknowledge more clearly that temporal and immediate context effects have been robustly (and independently) demonstrated in previous studies. Also, our new and shorter results section on the distractor effect, further highlights that our design can gauge the relevant contribution of the immediate and temporal context.

3. The evidence for a context effect in choice was rather weak (anecdotal evidence for a positive distractor effect). Some of this could be due to being underpowered. However, I wonder if a larger context effect would be observed using numerical rewards (instead of dots; see next point) and complete feedback instead of partial feedback? Previous research has found that context effects in these types of tasks are generally more pronounced with complete feedback (Bavard et al., 2018; 2021). I think running an additional experiment with these parameters, if possible, might help to bolster the overall conclusions.

References:

Bavard, S., Lebreton, M., Khamassi, M., Coricelli, G., & Palminteri, S. (2018). Reference-point centering and range-adaptation enhance human reinforcement learning at the cost of irrational preferences. Nature Communications, 9, 4503.

Bavard, S., Rustichini, A., & Palminteri, S. (2021). Two sides of the same coin: Beneficial and detrimental consequences of range adaptation in human reinforcement learning. Science Advances, 7, eabe0340.

Thank you for this insightful comment. We agree that numerically displayed reward and complete feedback could yield different context effects. Due to limited resources, we opted for running only one larger replication study with the same parameters as our original study, which we thought was imperative given the anecdotal nature of some of the original effects. We summarize the new results and associated changes in the “Overview of major revisions” section in this letter. However, we appreciate your suggestion about exploring additional task parameters and believe that this could be a very interesting future endeavor. Thus, we addressed this point on task parameters briefly in the Discussion.

4. The rationale for using moving dots instead of numerical values as rewards should be explained, given that this diverges from most other studies in this area.

Thank you for this comment. Please also see our reply to Reviewer's 1 first comment. In sum, the rationale for using moving dots was to emulate the internal uncertainty associated with the primary reward (drops of juice) consumption in the Louie et al (Louie et al., 2013). study. Reshuffling the frames of dots from trial to trial and presenting them briefly aimed to induce internal perceptual uncertainty.

5. Can the authors discuss whether the memory process assumed by the rank-based model occurs at the feedback encoding stage or the choice stage? The paper seems to imply that it occurs at the time of choice; however, this would not be able to explain the kinds of irrational transfer preferences that have been observed in prior studies and described in point (2) above. That is, participants in those studies sometimes form a strong preference for A over B, even though all of A's rewards were smaller than B's rewards. The only way this could occur is if the subjective values of the rewards were distorted by the local context at encoding time.

Thank you for this important comment. Given the absence of an immediate context effects in the choice phase (i.e., indistinguishable behaviour between binary and ternary trials), we maintain that the major contextual distortions had occurred already by the end of the learning phase and were revealed in the estimation task. That means, that the rank-based model is not at play during the choice. Indeed, for all models we assume that the values are distorted before the choice phase, and choices are determined through a probit function (we punctuate this more extensively when we describe the models in the main text). Thus our interpretation, which we hope that the revision clarifies, is consistent with the irrational transfer results. We also downplayed the involvement of memory as this can be definitively tested with our paradigm. The binary comparisons the RB model implies could happen either "online" (from one learning trial to the next) or once participants are prompted to provide their value estimates.

Minor comments:

1. Shouldn't the scale of the value representations in Figure 1A (y-axis) be between 0 and 1 for divisive and range normalization?

Not necessarily, as this depends on the formulation of the range and the divisive normalization. If we use the simplest possible divisive normalization version (without any other parameters like gain, saturation, normalization weight, ...) and the range model as depicted, then transformed values should range from 0 to 1.

In the paper we used the augmented version of these two models, which can yield values between 0 and any value, depending on the parameters. In the Materials and Methods section, we denoted the equations of the normalization models, as well as the parameters we used for the simulation, under "Model Simulation".

We would clarify that the version of the divisive normalization was used was the one in Louie et al. (Louie et al., 2013) given that our study aimed to assess the replicability of theirs.

Divisive normalization	$SV_i = V_i / \sum_{j=1}^n V_j$
Range normalization	$SV_i = V_i - \min(V_{j,j=1:n}) / \max(V_{j,j=1:n}) - \min(V_{j,j=1:n})$

2. Is Figure 2C how the estimation trials actually appeared? I was confused by the fact that in the figure, the scale goes from 4 dots, to 6, to 15 with equal spacing.

In Figure 2C we show a schematic (cartoon) of an estimation trial and also of the reward presentation. The idea was to show how the scale changes as participants use the slider. In an actual trial, participants saw a cloud of dots on the left representing 15 dots as the minimum reward and on the right a cloud of 85 dots as the maximum reward. In the middle, they could see the number of dots corresponding to the location of the slider (i.e., their estimate) as they were moving the slider in real-time. We have explained this in detail in the supplemental Material and now refer to it in the Figure caption.

3. P. 8, line 170: “using a two-sided t-test on.” This sentence may have been cut off.

This has been corrected.

4. Very minor, but I don’t think the units (ms) need to be in parentheses when they appear in confidence intervals. Removing the parentheses may make it less crowded.

This has been now fixed.

5. What should we make of the general tendency to overestimate the values of options (Fig. 3A)?

We believe that the overestimation occurred because we presented the dots in successive frames (on the motivation of using moving dots, please also see the Methods section, our reply to the 4th major comment and to reviewer’s 1 first comment). Indeed, when we performed the task ourselves, we do get a subjective feeling that the dots were more numerous than they were. Other mechanisms, such as perceptual aftereffects, and attention to extremes (e.g., frames with dots placement that made them appear more numerous) may have contributed to this general overestimation. We think that the overestimation can be attributed to the moving nature of the stimulus, and the motion-induced overestimation phenomenon discussed here (Afriz et al., 2004). In the main text we emphasized that the estimated values are strongly correlated with the actual values.

References

- Afraz, S.-R., Kiani, R., Vaziri-Pashkam, M., & Esteky, H. (2004). Motion-induced overestimation of the number of items in a display. *Perception, 33*(8), 915–925. <https://doi.org/10.1068/p5296>
- Baker, D. H., Vilidaite, G., Lygo, F. A., Smith, A. K., Flack, T. R., Gouws, A. D., & Andrews, T. J. (2021). Power contours: Optimising sample size and precision in experimental psychology and human neuroscience. *Psychological Methods, 26*(3), 295–314. <https://doi.org/10.1037/met0000337>
- Gold, J. I., & Shadlen, M. N. (2007). The neural basis of decision making. *Annual Review of Neuroscience, 30*, 535–574. <https://doi.org/10.1146/annurev.neuro.29.051605.113038>
- Louie, K., Khaw, M. W., & Glimcher, P. W. (2013). Normalization is a general neural mechanism for context-dependent decision making. *Proceedings of the National Academy of Sciences, 110*(15), 6139–6144. <https://doi.org/10.1073/pnas.1217854110>
- Smith, P. L., & Little, D. R. (2018). Small is beautiful: In defense of the small-N design. *Psychonomic Bulletin & Review, 25*(6), 2083–2101. <https://doi.org/10.3758/s13423-018-1451-8>